# Characterizing the Sea-Ice Floe Size Distribution in the Canada Basin from High-Resolution Optical Satellite Imagery

Alexis A. Denton[1], Mary-Louise Timmermans[1]

[1]Department of Earth and Planetary Sciences, Yale University, New Haven, 06511, USA

*Correspondence to*: Alexis A. Denton (alexis.denton@yale.edu)

**Abstract.** The sea-ice floe size distribution (FSD) characterizes the sea-ice response to atmospheric and oceanic forcing and is important for understanding and modeling the evolving ice pack in a warming Arctic. FSDs are evaluated from 78 floe-segmented high-resolution (1-m) optical satellite images capturing a range of settings and sea-ice states during spring through fall from 1999 to 2014 in the Canada Basin. For any given image, the structure of the FSD is found to be sensitive to a
10 classification threshold value (i.e., to specify an image pixel as being either water or ice) used in image segmentation, and an approach to account for this sensitivity is presented. The FSDs are found to exhibit a single power-law regime between floe areas 50 m$^2$ and 5 km$^2$, characterized by exponents (slopes in log-log space) in the range -2.03 to -1.65. A distinct linear relationship between slopes and sea-ice concentrations is found, with steeper slopes (i.e., a larger proportion of smaller to larger floes) corresponding to lower sea-ice concentrations. Further, a seasonal variation in slopes is found for fixed sites in
the Canada Basin that undergo a seasonal cycle in sea-ice concentration, while sites with extensive sea-ice cover year-round do not exhibit any seasonal change in FSD properties. Our results suggest that sea-ice concentration should be considered in any characterization of a time-varying FSD (for use in sea-ice models, for example).

## 1 Introduction

The Arctic Ocean is covered perennially to varying extent by sea ice floating in discrete fragments called floes, which range
in size from O(1) m to O(100) km (Untersteiner, 1986). This assortment of sizes, which may be described by a sea-ice floe size distribution (FSD, see Rothrock and Thorndike, 1984) influences and is influenced by the ice pack response to thermal and dynamic atmospheric and oceanic forcing: for example, a distribution with a larger fraction of smaller, thinner floes will melt more rapidly (e.g., via lateral melting) (Steele, 1992), and deform and drift with less resistance than a field comprised of more larger floes. In turn, the FSD influences energetics and mixing in the upper ocean through a variety of processes, such as
spatially variable momentum transfer and buoyancy fluxes that generate small-scale ocean flows (e.g., Mensa and Timmermans, 2017; Smith et al., 2002). Bateson et al. (2020) account for varying floe sizes in a sea-ice model (developed for use in a climate model) via an FSD that is iteratively modified by melt/growth and dynamical processes; they demonstrate that melt patterns (e.g., basal vs. lateral melt) differ significantly when a size distribution is accounted for (see also Roach et al., 2018). Accurate observational characterization of the FSD yields insight into the physics of the ice cover and its surroundings

and provides validation of Arctic modeling studies which incorporate the FSD to more accurately represent these processes and their seasonality (e.g., Horvat and Tziperman, 2015; Roach et al., 2019; Zhang et al., 2015; Zhang et al., 2016).

The sea-ice FSD has been characterized extensively in observations since the seminal paper of Rothrock and Thorndike (1984); the FSD may be quantified in a number of ways, for example as the number of floes per unit area of the region in question which have sizes that are not smaller than a given size. In general, the FSD resembles a single power-law (e.g., Gherardi and Lagomarsino, 2015; Hwang et al., 2017; Stern et al., 2018b) or two distinct power laws depending on floe scales (Geise et al., 2017; Steer et al., 2008; Toyota et al., 2011; Toyota et al., 2006), although alternate distributions have been explored (see e.g., Herman, 2010, and the discussion by Stern et al., 2018a). There are limited FSD characterizations that span a comprehensive range of floe scales, from O(1) m to more than O(10) km (see Stern et al., 2018a). This is in part due to a reliance on high-resolution aerial photography with limited area coverage and sampling. While Stern et al. (2018b) find that a single-power law may describe the FSD across floe scales ranging from 10 m to 30 km, it remains an open question as to whether a single power law holds across all floe scales and in all settings, or whether there may be two distinct power-law regimes. The seasonal evolution of observed FSDs has been the subject of several recent observational studies (Hwang et al., 2017; Perovich and Jones, 2014; Stern et al., 2018b), each of which finds a steepening of the FSD slope into summer. This slope increase in the melt season is thought to be related to the break-up of floes beginning in the spring in tandem with melt through the summer reducing the proportion of larger to smaller floes (e.g., Stern et al., 2018b).

A collection of high-resolution optical satellite images, spanning nearly two decades, from different locations within the Canada Basin, allows us to test and refine previous findings for a variety of settings, and for floe sizes in the range of 5 m² to 100 km². In the next section we introduce the collection of images and describe our image segmentation methodology and FSD construction. In Sect. 3, we show how FSDs exhibit a single power-law behavior spanning the full range of floe sizes and provide evidence for a shoaling of the slope of the distribution (i.e., increased ratio of larger to smaller floes) as sea-ice concentration increases. This finding is consistent with a seasonal evolution of the FSD found here, which we describe in context with previous studies in Sect. 3.4. Results are summarized and discussed in Sect. 4.

## 2 Data and Methods

### 2.1 Satellite Imagery and Environmental Parameters

We perform a floe-size distribution analysis on 78 high-resolution, cloud-free, electro-optical satellite images of sea ice in the Canada Basin acquired from a United States military passive satellite sensor from 1999 through 2014 (excepting years 2003–2005 and 2009), declassified as a part of the military and scientific coalition MEDEA program (Baker and Zall, 2020), and distributed to the public through the United States Geological Survey (USGS) Global Fiducials Library (GFL) Program. The images were obtained during April through September of those years over various geographic locations (Fig. 1a and Table 1),

including three stationary "fiducial" sites in the Beaufort and Chukchi Seas, and the Northern Canada Basin, designated as consistent locations within the Basin for inter-annual comparison of environmental observations. The 2013 and 2014 image sets contain additional images acquired at non-fiducial sites over designated drifting floes and released through the GFL in support of the National Aeronautics and Space Administration Operation IceBridge, and the Office of Naval Research Seasonal Ice Zone Reconnaissance Surveys (SIZRS) and Marginal Ice Zone (MIZ) Departmental Research Initiative (DRI) field campaign (see Lee et al., 2012). The images are panchromatic (with uncalibrated grayscale pixel values ranging from 0 to 255) and projected onto the Universal Transverse Mercator (UTM) grid with a resolution of 1 m; the SIZRS images have a resolution of 1.3 m. The images cover areas O(1–1,000) km² and allow for characterization of the sea-ice FSD on scales from O(1) m² to O(100) km². We note that partially or fully cloud-covered images on the GFL were generally unambiguous and rejected outright from our analysis. Cloudy pixels either fully obscure information about the ice cover below or interfere with the proper identification of floe outlines. For further description of the MEDEA imagery see Kwok (2014) and Baker and Zall (2020).

We examine the FSD for all 78 images in the context of the following environmental parameters: sea ice concentration SIC (fractional area of sea ice in the image), distance to the ice edge (km), and surface air temperature (SAT, °C), Table A1. SIC is calculated for each image by dividing the total identified ice area (including that of border-intersecting floes) in the segmented image by the total area of the image. This is compared with SIC from passive microwave satellite data for the dates and locations of the images. Distance to the ice edge is computed as the distance (rounded to the nearest 100 km) between the image location and the nearest point on the median ice edge contour (defined where the concentration is 15%) for the month and year of the image. SIC from passive microwave data are from the National Oceanic and Atmospheric Administration/National Snow and Ice Data Center (NOAA/NSIDC) Climate Data Record of Passive Microwave Sea Ice Concentration, Version 4 (Peng et al., 2013; Meier et al., 2021). Median ice edge contours are from the NSIDC Sea Ice Index, Version 3, and are derived from passive microwave SIC data (Fetterer et al., 2017). SAT (at 2-m) is retrieved from the European Centre for Medium-Range Weather Forecasts (ECMWF) ERA5 Reanalysis (Hersbach et al., 2020) hourly data on single levels from 1979 to present (Hersbach et al., 2018), and taken as the mean daily value for each image region on the corresponding image day.

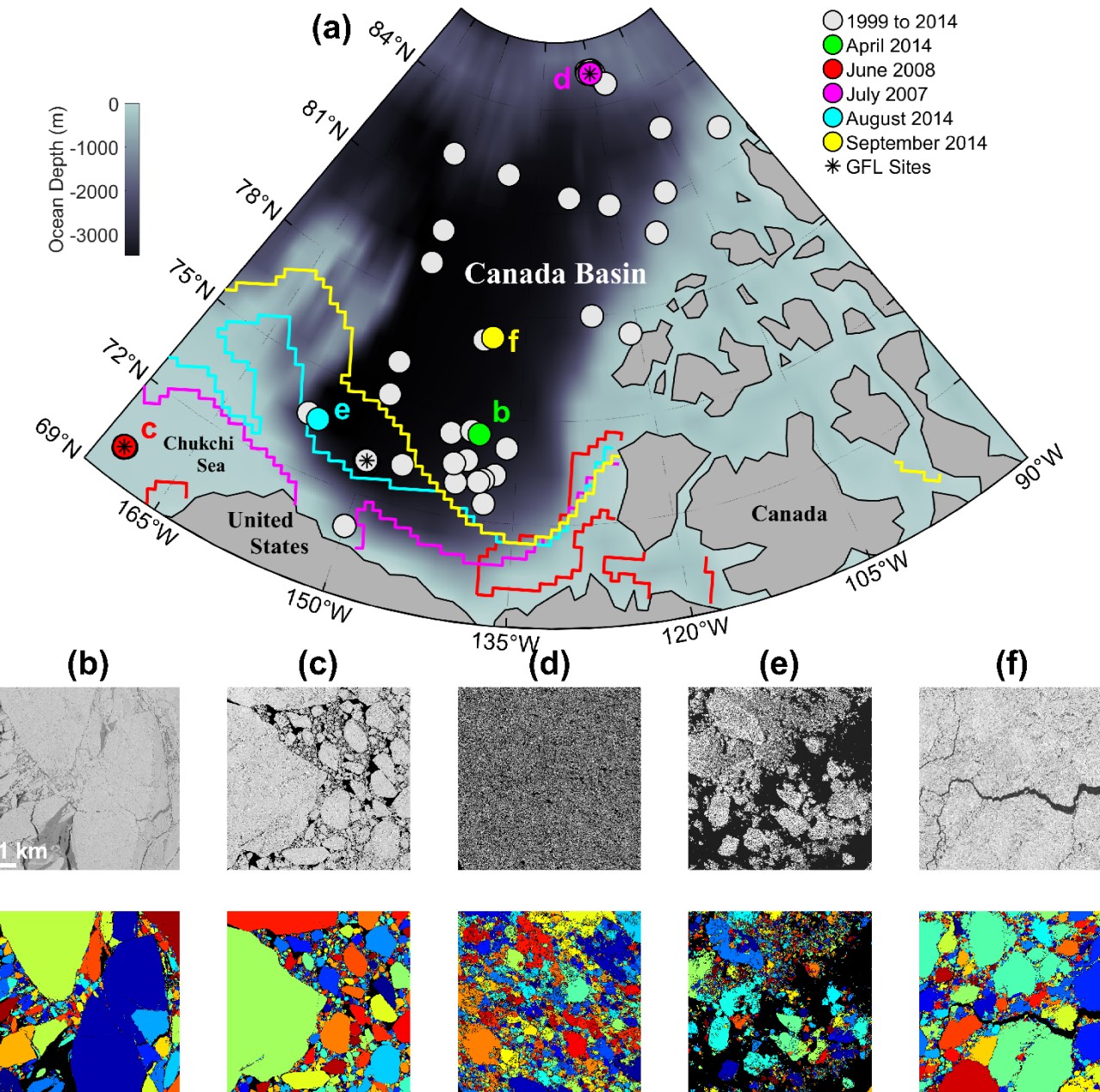

**Figure 1. Map of study region with image locations, and example subsets of images and corresponding segmentations. a.** Study region within the Canada Basin with locations of 78 images from 1999 to 2014 (gray circles). **b. through f.** 100 km$^2$ image subregions (*top*) and corresponding image segmentations (*bottom*) from b. 30 April 2014, c. 12 June 2008, d. 23 July 2007, e. 11 August 2014, and f. 20 September 2014. Locations of images b (green)–f (yellow) are labeled on the map. USGS fiducial sites (black asterisks), for which there are images from multiple years, are noted to the southeast of e (Beaufort Sea), at c (Chukchi Sea), and at d (northern Canada Basin). Median ice extents (bounding the area with more than 15-percent concentration) are shown for b–f in corresponding colored lines for those months (April 2014 extent is south of the map domain). The median monthly ice extents are from the NSIDC Sea Ice Index, Version 3 (Fetterer et al., 2017).

|  | Beaufort | Chukchi | N. Canada Basin | Other | Total |
|---|---|---|---|---|---|
| **April** | 5* | 2 | 1 | 17 | 25 |
| **May** | 7* | 2 | 7* | 4 | 20 |
| **June** | 2* | 5* | 1 | 7 | 15 |
| **July** | 1 | 1 | 3* | 3 | 8 |
| **August** | 1 | 0 | 3* | 2 | 6 |
| **September** | 1 | 0 | 1 | 2 | 4 |
| **Total** | 17 | 10 | 16 | 35 | 78 |

Table 1. Number of Images acquired at the Beaufort, Chukchi, and Northern Canada Basin fixed USGS fiducial sites (designated in Fig. 1a by asterisks); at GFL locations corresponding to other programs (including NASA's Operation IceBridge, and ONR's SIZRS and MIZ DRI, designated in Fig. 1a by light gray circles at locations with no asterisks); and in total. Images acquired on the same day at the same site are not independent samples; their presence is denoted by *.

## 2.2 Image Segmentation

An algorithm for segmentation of individual sea-ice floes in the images is developed (Denton, 2022), using a combination of "restricted growing" steps (Soh et al., 1998), with the addition of an alternative approach (described in Sect. 2.2.1) to the first step of the algorithm, which requires the image to be preprocessed into a binary image. Generally, each image is first manually classified into ice (floes) and water (background) separated by some grayscale threshold based upon the image pixel value histogram in which low grayscale values indicate dark open water and high values indicate bright ice. The classified image is then segmented via an iterative erosion-expansion scheme in which floe-edge pixels are converted to water pixels via binary filter (see Soh et al., 1998) until a distinct separation of individual floes is apparent (via visual check). The eroded and separated floes are then individually labeled and subsequently expanded to their original size (see Paget et al., 2001). Only the largest floes are segmented and their ice pixels removed from the binary image after the first erosion-expansion iteration, and the binary image is subsequently eroded iteratively to lesser degrees to separate the remaining smaller, unsegmented floes (see Stern et al., 2018b). Finally, any floes cut off by the image borders are removed. Floe areas are retrieved from the segmented image to construct an FSD, described in Sect. 2.3. We limit our FSD analysis to floes having an area of at least 5 pixels, or 5 m² (smaller scales are indistinguishable from noise) and consider floe areas over the range of 5 m² to 100 km².

There are two main steps in erosion-expansion segmentation which require a choice of parameter at the discretion of the user: classification and erosion.

*Classification: Choice of Grayscale Threshold*

Classification separates ice pixels from ocean pixels via the choice of a threshold grayscale value. A grayscale optical satellite image of sea ice ideally contains two peaks in its histogram: a bright-ice peak nearer to values of 255 and a dark-ocean peak

nearer to values of 0 (see Fig. 2g; note that pixel values have been scaled to fall between 0 and 1). The threshold must fall between the histogram ice and water peaks to separate ice floes from ocean. This choice of the precise threshold (see Sect. 2.2.1) can be made difficult by the distance between the histogram peaks being large (as in Fig. 2g), the peaks being flattened or nonexistent, or the presence of a third peak or cluster of peaks between the ocean and bright-ice peaks, resulting from classes which are not easily categorized as ice or ocean (e.g., thin, dark ice, or melt ponds, or ridge shadows).

*Erosion and Expansion*

Erosion converts any ice pixels adjacent to ocean pixels in the classified image into ocean pixels. This has the visual effect to erode the ice floes away from each other, but also to expand any clusters of ocean pixels in floe interiors (e.g., melt ponds classified as ocean), possibly leading to division of a single floe into multiple floes. Erosion is done iteratively enough times to provide full separation of floes, with clear boundaries of ocean between them. The eroded binary image is then filtered in a process called filling, in which any ocean pixels in the interior of individual floes are converted to ice pixels (see Stern et al., 2018b); this has the effect visually to fill ocean holes in floes, and practically to suppress artifact floes from emerging in floe interiors during the subsequent expansion step. The eroded filled floes are then labeled with a unique positive integer value.

The binary image is then filtered one last time in a process called expansion, in which eroded pixel bands around floe edges (ocean pixels which were originally ice pixels) are converted back to ice pixels, band by band the same number of times as the number of erosions. At every step, these pixels are assigned a value equivalent to the positive integer mode of the surrounding 8 pixels (or a new unique positive integer value if all neighboring pixels represent ocean and have a value of 0), until all floes are expanded to their original size with unique numerical labels (see Paget et al., 2001). This process is repeated hierarchically in which the largest floes are segmented and removed from the binary image first, and the smallest floes are segmented last; this is because the number of erosions required to separate the largest floes will also completely erode the smallest floes, leaving them unlabeled in the segmented image (see Stern et al., 2018b). The number of erosions at each hierarchical step is chosen such that floe separation is maximized while expansion of ocean holes within floes is minimized.

### 2.2.1 Selection of Classification Threshold

Past segmentation studies have chosen a classification threshold that reduces features on the surface of a floe (e.g., melt ponds or ridge shadows classified as ocean; see Paget et al., 2001, their Fig. 1a; and Stern et al., 2018b, their Fig. 3b). The motivation for this choice, which is a lower threshold value, is to avoid the growth of ocean holes in a floe and to reduce artifact floes (see e.g., Paget et al., 2001). On the other hand, small floes (having horizontal scales less than around 40 m) do not separate well after classification with a lower-threshold approach because the grayscale pixel values of the boundaries of small floes tend to be similar to that of surface features (see Fig. 2); small-floe boundaries will be ill-defined in the classified image, and they may be assigned as belonging to larger floes. Further, artifact floes emerge where small floes are ill-defined or where a few

surface features have survived low-threshold classification. These artifact floes are apparent by their rectangular edges which result from the row-by-row sweep across the image of the erosion-expansion filters; if usually rounded floe edges or surface features are not well-defined, the effect of the filters will be to impose linear edges and features.

To alleviate the issues described above, we take an alternative approach and choose a threshold value that is sufficiently large that small-floe (horizontal scales less than around 40 m) boundaries are well-defined in the binary image (Fig. 2e). Large floes remain well-defined, even if the larger threshold results in ocean pixels within their interiors. The number of erosions required to properly separate floes at each hierarchical step is much fewer [O(1) compared to O(10), see e.g., Paget et al., 2001; Stern et al., 2018b] if a larger threshold is chosen. Performing fewer erosions limits the expansion of floe-interior ocean holes and

the emergence of artificial floes. The filling step also acts to alleviate emergence of artifact floes. Examination of the histogram of pixel grayscale values for any given image suggests a natural choice of threshold as the local minimum between two local maxima (dark ice/melt ponds and bright ice). In practice, the choice must usually be made using iterative adjustments to this location after visual checks of the classified image; here, we iteratively increase the threshold above the minimum until the edges of small floes are appropriately delineated (see Fig. 2).

This choice of higher threshold yields adequate identification of smaller floes in the image, with smaller and fewer artifact floes (those with rectangular edges, see Fig. 2f compared with Fig. 2b). A secondary benefit of the high-threshold approach presented here is speed. The expansion step occupies the most time (mode filtering is a computationally intensive process); because the number of expansions will match the number of erosions, reducing the number of erosions by an order of

magnitude will significantly speed up the segmentation. In practice, we find that using the low-threshold approach of Paget et al. (2001) on our dataset results in a segmentation time of O(10) minutes to O(1) days, while using our high-threshold approach results in a segmentation time of O(1) minutes to O(1) hours.

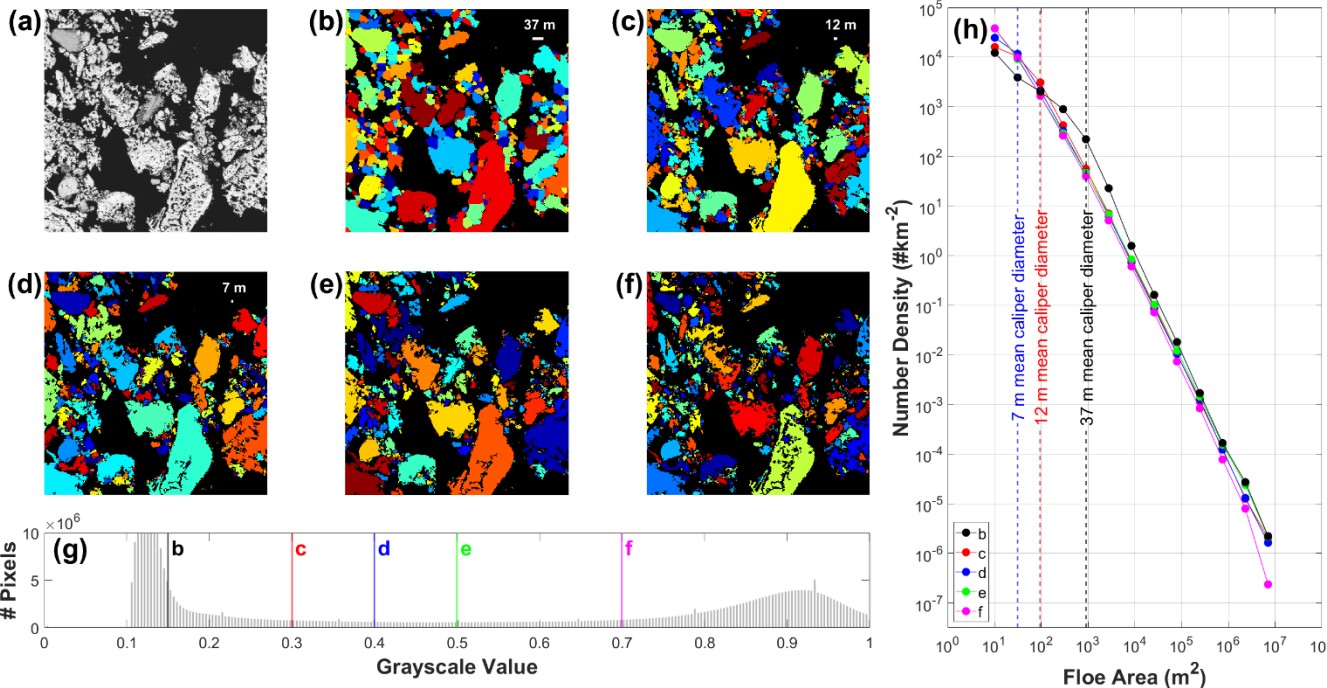

Figure 2. Comparison at the small-floe scale of segmentation of an MIZ MEDEA image from 11 August 2014 using different parameters. a. A 750 m × 750 m subregion of the image showing heavily ponded, broken ice and open water; b. through f. segmentations of the same subregion obtained by applying grayscale thresholds (on a scale of 0 to 1) of 0.15, 0.3, 0.4, 0.5, and 0.7, respectively; g. histogram of pixel grayscale values (scaled from 0–255 to 0–1) for the overall image showing the location of grayscale thresholds for b (black) through f (magenta); and h. floe size distributions of b (black) through f (magenta). Vertical lines are shown at scales shown in b through d which correspond to the size of artifact floes in each.

## 2.3 Floe Size Distribution

We construct the FSD using a number density $n(a)$, computed as the fractional number of floes in the scene having area between $a$ and $a + da$, divided by the width of the bin, $da$. We use 15 bins spaced logarithmically (such that bin sizes increase with larger areas) from 5 m$^2$ to 100 km$^2$, with a minimum floe number requirement of 2 per bin. If the FSD satisfies a power-law, the number density will fall along a straight line in a log-log plot; we can write $n(a) = ca^m$ for $0 < a < \infty$ where $c$ is a normalization constant, and the distribution has slope $m$. We test the sensitivity of the FSD to the choice of bin number by varying the number of bins from 15 to 5 and find that the shape of the FSD is stable between 10 and 15 bins. Due to the sparsity of floes in the largest bins, a result of the finite size limit of whole large floes being captured in satellite images, we limit the linear fit in log-log plots (to estimate $m$) to floe areas smaller than 5 km$^2$.

The FSD defined above is a non-cumulative form, while some studies present the cumulative form of the FSD (i.e., the integral of the probability density function). When the non-cumulative FSD is a straight line on a log-log plot, its cumulative form will not be a straight line when the maximum floe size has some finite upper bound. Rather, the cumulative FSD in log-log space will be concave down (see the discussion by Stern et al., 2018a). The cumulative FSD may present both a flattened slope over

small-floe scales and a steep slope in the large-floe tail (e.g., Hwang et al., 2017, Figure 1d), neither of which can be discerned as purely physical. Interpretation of the cumulative FSD is ambiguous because this concave-down behavior may alternatively be a manifestation of the distribution of ice floe sizes having multiple power-law regimes.

The floe size may be taken to be floe area $a$, perimeter, or a diameter proxy such as mean caliper diameter (MCD), used commonly after Rothrock and Thorndike (1984). In the present work, we use floe area because we obtain this directly in the segmentation (and it is directly relatable to floe models), although this is easily related to the MCD (see Rothrock and Thorndike, 1984; Stern et al., 2018a). We note however, that relating FSDs derived from $a$ and MCD requires caution (see Sect. 3.4).

### 2.3.1 FSD Sensitivity to the Choice of Classification Threshold

The size of the "artifact" floes discussed in Sect. 2.2.1 (where the size is shown in scale bars on Fig. 2b–d), corresponds to the scale of an apparent change in slope of the corresponding FSD (Fig. 2h, black, red, and blue dashed lines), in which the slope (exponent) is steeper for floe areas larger than this scale and flatter for floe areas smaller. This appears to result from an over-identification of floes at the artifact scale, and an under-identification of the floes smaller than it. Testing a range of classification thresholds shows how the scale of artifact floes is affected by this choice, as is the resulting scale at which there is a change in FSD slope (Fig. 2b through d): a higher threshold choice eliminates the spurious change in FSD slope caused by such floe mis-identifications around this artifact scale.

A potentially undesirable effect of a high threshold is that larger floes may be incorrectly divided into multiple floes (see Fig. 2c through f). However, such over-segmentation of larger floes seems to have a minimal effect on the slope of the FSD for floe areas larger than the artifact scale (as in Fig. 2h). At the small-floe scale, the lower-limit to reduction of the size of artifact floes through adequate segmentation is around 50 m$^2$ for our image dataset. For this and the reasons described above, the slope of the FSD is valid only for floe areas larger than 50 m$^2$ and we limit fitting in log-log plots (to estimate $m$) to floe areas between 50 m$^2$ and 5 km$^2$ (see also Sect 2.3). The number of floes which fall in this fitting range (see Table A1) is between 10 to 55 percent of the total number of whole floes segmented across the entire floe range between 5 m$^2$ and 100 km$^2$ (with the minimum floe count requirement of 2 per bin). Any segmentation resulting from this approach which identifies floes inadequately or cannot be validated due to visual ambiguity of the ice field is not included for analysis. Certain segmentations which, upon visual validation, are neither wholly adequate nor inadequate are retained for analysis but are tagged in plots in the results and are indicated in Table A1 as low confidence.

### 2.3.2 FSD Power-Law Fit Evaluation

The maximum likelihood estimator (MLE, see Clauset et al., 2009) can be preferable for the determination of FSD slopes as it does not rely on specifying bins or fitting ranges (Hwang et al., 2017; Stern et al., 2018a; Stern et al., 2018b). In addition to

least-squares fitted slopes $m$ and following Clauset et al. (2009), we compute FSD MLE slopes $m_{MLE}$, and conduct goodness-of-fit tests on these power-law fits, reporting corresponding $p$-values (where the $p$-value is the probability that the difference between the model fit and the observed FSD could be due to statistical fluctuations; see Clauset et al., 2009 for a detailed discussion). The power-law fit is a plausible model for the FSD if the computed $p$-value is sufficiently large ($p \geq 0.1$); otherwise, the power-law model must be rejected.

Clauset et al. (2009) argue that a strict statistical lower bound on power-law behavior must be computed for the observed distribution; we compute these values $a_{min}$, following their methodology. Because $m_{MLE}$ and $a_{min}$ are determined directly from the unbinned floe areas for each image, we compute both over all floe areas ($\geq 5 \text{ m}^2$), and do not exclude floes at or below the artifact scale.

## 3 Results

### 3.1 FSD Slope Characteristics

Results indicate that FSDs are characterized by a single power law with (linear least-squares fit) slope $m$ for the entire regime of floe areas between 50 m$^2$ and 5 km$^2$ (Fig. 3a). Slope values $m$ range from -2.03 to -1.65 (Table A1) with a mean across all images of $-1.79 \pm 0.08$. This single power-law structure is consistent across all images (Fig. 3a), which span six months from initial spring break-up in April to the September ice minimum for a fifteen-year period, and a range of sea-ice settings from the MIZ to the interior pack.

We find no significant difference between slopes $m$ and $m_{MLE}$ (Table A1). The mean $m_{MLE}$ over all images is $-1.77 \pm 0.11$. Considering each image, $m_{MLE}$ differs from $m$ by about 3% on average. We find that 76% of the fits pass the goodness-of-fit test with $p \geq 0.1$ (Table A1) meaning that the FSDs can plausibly be power-law distributed. Finally, we find that the strict lower-bound to power-law behavior $a_{min}$ varies considerably over the images (Table A1), spanning around 10 to 10,000 m$^2$ with a median value of 361 m$^2$. Considering that the largest floe areas in the images are around 10 to 100 km$^2$, the range of floe sizes over which the power-law fits apply is large. Values $a_{min}$ can vary significantly even across images acquired on the same day at the same location (see e.g. Table A1, images 14–15).

Examining $m$ from 1999 to 2014 reveals that there is no apparent overall interannual trend of FSD slopes in the Canada Basin. It might have been expected that a steepening of the slope (i.e., a higher portion of small to large floes, and more negative FSD slopes) over multiple years would occur as the sea-ice thins and summer concentrations decline. However, we find no evidence for an overall change in $m$. It may be that the latitudinal span of images obscures any temporal variability over the 15 years analyzed.

Partitioning the image FSD slopes by month, we find that there is no apparent variation in *m* with season (Fig. 3b). In the next
section, we consider FSD slopes retrieved at the three fixed GFL fiducial sites (see Fig. 1a) to investigate whether there may
be a seasonal signal obscured by spatial variability of the sample locations. There is an increasing spread in the values of *m* in
any single month as the season progresses from April through September. We will show that in these later months, the broad
latitudinal distribution in images is accompanied by a significant latitudinal distribution of SICs and SATs. We further note
that we have low confidence in some segmentations in late summer months (those shown by gray dots in Fig. 3b–d);
appropriate segmentation of images in which the effects of melt are prominent (e.g., extensive ponding and slush ice) can be
problematic, especially when validation by eye is not possible.

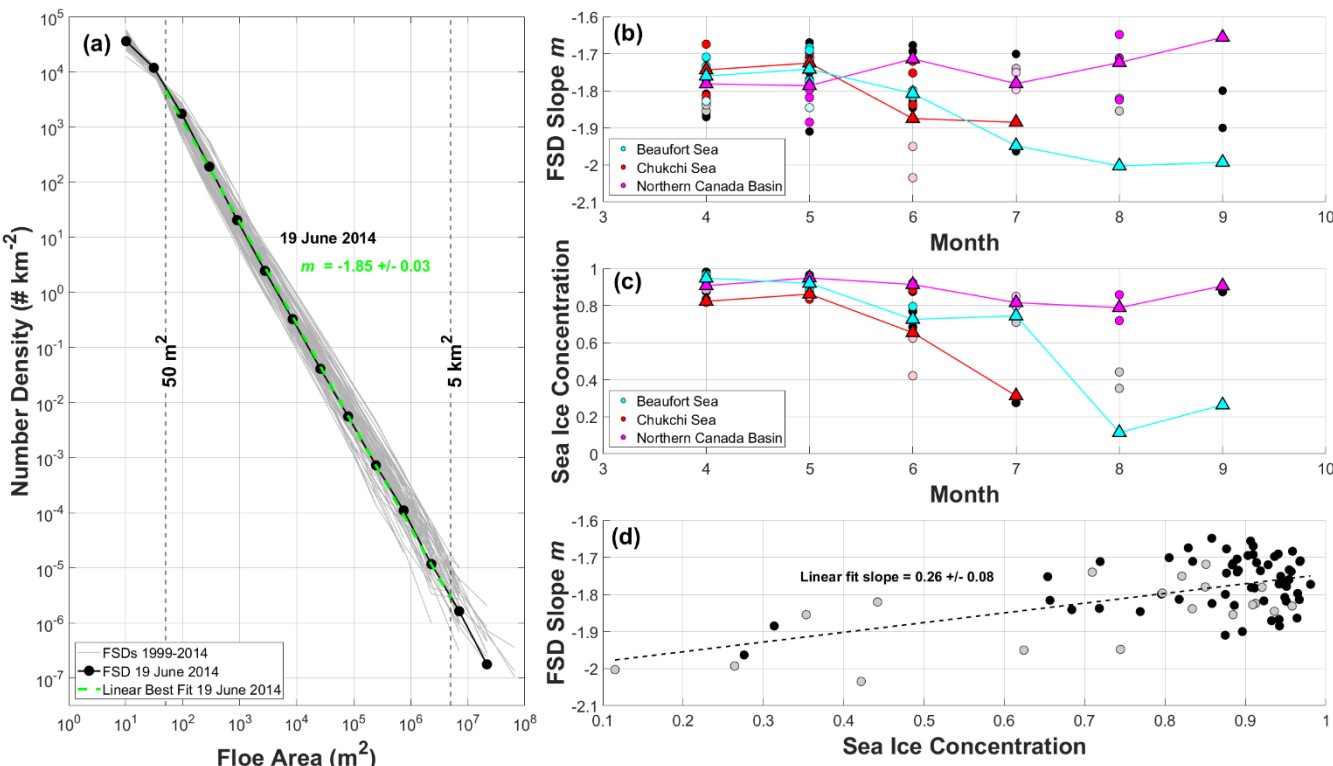

**Figure 3. FSDs and FSD slopes *m* versus month, SIC (fractional area) versus month, and slopes *m* versus SIC, for 78 satellite images acquired from 1999 to 2014 in April through September of those years. a. FSDs (gray lines) are plotted on a log-log scale using 15 logarithmically spaced bins for the range of floe areas spanning 5 m² to 100 km² and the requirement of a bin count of at least 2 floes. A representative FSD is shown for the 19 June 2014 image (black line with solid circles) with a linear best-fit (green dashed line) and slope *m* (green). Fits are taken from 50 m² to 5 km² for reasons discussed in Sects. 2.2.1 and 2.3. b. FSD slopes *m* versus month. In b–d, black dots are shown for images segmented with high-confidence and gray dots for those segmented with low-confidence. Slopes *m* for the three GFL site (see Fig. 1a) images only are shown in cyan (Beaufort Sea), red (Chukchi Sea), and magenta (Northern Canada Basin), with mean monthly slopes (triangles). c. SIC versus month. SICs for the GFL site images only are shown again in cyan (Beaufort Sea), red (Chukchi Sea), and magenta (Northern Canada Basin), with mean monthly SICs (triangles). In b and c, individual slopes and SICs from low-confidence segmentations at the GFL sites are shown in a lighter shade of each site's designated fill-color. d. FSD slopes *m* versus SIC and linear fit (black dashed line) with slope (and 95% confidence intervals).**

## 3.2 Seasonal Variability at Stationary Locations in the Canada Basin

Considering only the 17 images at the Beaufort Sea site (73°N, 150°W), which span the whole range of years and months, we find that a clear seasonal signal in slope emerges (Fig. 3b, cyan line). The mean slope $m$ at the Beaufort site is shallowest in April and May, and then steepens through August, increasing only slightly through September (only a single image is available for each month from July to September at the Beaufort site).

At the Chukchi site (70°N, 170°W), 10 images span years 2006–2014 and only for months April through July. While we cannot examine the entire spring–fall seasonality of Chukchi FSD slopes, there is evidence of a similar start to the seasonal signal as that of the Beaufort, with greater variability in April and June. In April and May, mean $m$ at the Chukchi site is shallowest, steepening for June and July (Fig. 3b, red line).

Examining $m$ from 16 images spanning years 2000–2014 and the entire range of months at the northern Canada Basin site (85°N, 120°W), we find no discernable seasonal variability of FSD slopes (Fig. 3b, magenta line). We posit that a lack of seasonal signal in the northern Canada Basin is due to the lack of a seasonal signal in SIC at that location, which we discuss in the next section.

Note that for each of the GFL sites, we compute mean monthly slopes after first taking the mean of any slopes from images acquired on the same day at a particular site, to account for the fact that these are not independent estimates (see Table 1). We do the same for mean monthly SIC at the GFL sites, discussed in the next section.

## 3.3 Relationship between FSD Slope, Sea Ice Concentration, and Surface Air Temperatures

It is notable that seasonal variations in SIC are only apparent for the images at the Beaufort and Chukchi sites, and not at the Northern Canada Basin site (Fig. 3c). At both the Beaufort and Chukchi sites, the evolution of monthly mean SIC (highest in April and May and decreasing through the summer) closely resembles the seasonal evolution of $m$ for the sites. Mean SIC at the Northern Canada Basin site exhibits virtually no seasonality, and always remains above 0.80, in the same way that $m$ does not vary much from spring to fall at that site.

There is a statistically significant linear relationship between $m$ and SIC (Fig. 3d), with $m$ shoaling as SIC increases. The best-fit linear slope is the same (within 95% uncertainty) if values of $m$ for segmentations with poor confidence (gray dots) are excluded from the fit. Note that there are more sample points in the high SIC range than in the low range, and the linear fit can only explain 33% of the variation (r-squared is 0.33) in $m$ with SIC. However, the linear relationship is statistically significant with a $p$-value of $O(10^{-8})$ (i.e., < 0.01).

The FSD may logically be expected to differ with distance to the ice edge if, for example, wave propagation into the ice pack plays some role in governing floe break up (see discussion in Toyota et al., 2011; Toyota et al., 2016). In the set of images analyzed here, the variation in $m$ with distance to the ice edge (not shown) is not straight-forward. For those images with SIC less than 0.80, which range from 0 to 1600 km from the ice edge, $m$ appears to generally shoal with increasing distance from the ice edge. However, for images with SIC greater than or equal to 0.80, which range from 200 to 3600 km to the ice edge, $m$ exhibits no clear variation with distance to the ice edge. SIC is not linear with distance to the ice edge at the location of images analyzed here; any tie between distance to the ice edge and $m$ is likely dominated by SIC.

With respect to SAT, we find that $m$ is relatively constant (between around -1.9 to -1.7) for a large range of temperatures (mean SAT over the day of a given image), in the range -25 to -2°C, with no statistically significant linear relationship between $m$ and SAT. In a "melt" regime (which we define to correspond to SATs between -2 and 4°C), $m$ values span their entire range (between around -2.0 and -1.6). This shows again the increased range of FSD slopes during the warmer months. Considering only the Beaufort and Chukchi Sea GFL sites reveals a similar structure to the two overall temperature regimes for FSD slopes: a cold regime in which values of $m$ remain relatively constant, and a melt regime in which values of $m$ span nearly their entire range. At the Northern Canada Basin site, on the other hand, SATs remain predominately below 0°C and $m$ remains shallow in the melt regime between temperatures -2 and 0°C (i.e., there are no $m$ values <-1.82). Finally, we note that for this image set, there is no clear relationship between SIC and SAT, again because for the melt range of SATs, SICs span their entire range. That is, the SIC relates directly to the FSD slope $m$, while there is no relationship between SAT and $m$.

### 3.4 Context with Previous Studies

It is useful to compare our slopes to relevant previous studies in the same region (Stern et al., 2018b; Stern et al., 2018a; Hwang et al., 2017). Stern et al. (2018b) examined the non-cumulative FSD using MCD, $x$, and plotting a floe number density $n(x)$, where $a \sim x^2$. From an application of basic probability theory, slopes reported in studies that examine the non-cumulative FSD using normalized floe number densities constructed from $x$ are equivalent to $2m + 1$ (where $m$ refers to slopes found in this study). Note that this is not the same for comparison to slopes of the cumulative FSD (see Stern et al., 2018a, their Table 1 footnotes).

Stern et al. (2018b) analyzed moderate-resolution (250-m) satellite images and characterized the FSD in the Beaufort and Chukchi seas during the summers of 2013 and 2014, finding that a single power law describes the FSD across floe diameters 2 to 30 km. Applying the transformation above to the reported range of slopes in Stern et al. (2018b) (-2.81 to -1.90, their Table 4) yields -1.91 to -1.45, which overlaps closely with the range of $m$ found here. We do not expect complete overlap of our slope range with theirs as they report mean monthly slope values, whereas our range is reported for the entirety of segmented images. We note that for our analysis of the same subset of an image analyzed by Stern et al. (2018b) (8 July 2014,

their Fig. 10; image not included in our analysis due to partial cloud-cover), our segmentation characterized by FSD slope $m$ agrees exactly with theirs (upon applying the transformation).

Stern et al. (2018b) additionally analyzed the FSD in 12 subregions of 3 high-resolution MEDEA images in 2014 in the Beaufort Sea and concluded that a single-power law characterization may extend to floe scales as small as 10 m, although the authors note that this conclusion is only supported by visual comparison of the FSD slopes on the smaller scale and those on the larger scale (from the moderate-resolution images), and not from statistical, quantitative comparison. Here, we extend the study of the small-scale behavior of the FSD from 3 high-resolution images over one summer to 78 over twelve summers in the same Arctic region and surrounding it, and find that a single power law is indeed applicable to the FSD across floe areas of 50 m$^2$ to 5 km$^2$, equivalent to a floe diameter range of ~ 9 m to 3 km (using the area to MCD relation in Rothrock and Thorndike, 1984, $a = 0.66x^2$).

With respect to seasonal variability, Stern et al. (2018b) found similar seasonal variations for floe sizes in the 2- to 30-km range in Beaufort and Chukchi FSD slopes (steepening from April through August and shoaling again in September). They point out that this is consistent with spring through summer break-up of larger floes, the shrinking of floes due to summer melt, followed by removal of the smallest floes at the end of melt and fall freeze-up of the ice field into large floes again. Hwang et al. (2017) examined the cumulative FSD for floe MCDs larger than about 100 m using TerraSAR-X Synthetic Aperture Radar images from 2014 in the Beaufort Sea region. They relate floe fracturing and corresponding steepening in FSD slopes (over a similar range of scales described by $m$) to a sequence of wind-driven deformation events over one summer season in the Beaufort Sea. They demonstrate a distinct steepening of the FSD slope in August which they relate to the timing of melt becoming dominant.

## 4 Summary and Discussion

We have segmented and retrieved the areas of Arctic sea-ice floes from 78 high-resolution optical satellite images acquired in the Canada Basin between 1999 and 2014. Our analysis of the resulting FSDs shows that the distributions exhibit a single power-law behavior across floes ranging in area from 50 m$^2$ to 5 km$^2$. We find that the slope $m$ of the power-law in log-log space ranges from -2.03 to -1.65 and shoals with increasing SIC. We find that, correspondingly, at locations within the Canada Basin which experience a distinct reduction in SIC from April through August and an increase in September, a similar seasonal signal in $m$ appears. On the other hand, at locations which undergo no distinct change in SIC through the summer, $m$ remains constant.

While we might have anticipated that any seasonality in $m$ might be related to seasonal changes in SAT, consistent with melt onset (see e.g., Hwang et al., 2017; Stern et al., 2018b), we find that seasonal variation in $m$ is more directly related to changes

in SIC. These findings provide support for an approach that uses SIC in any characterization of the FSD. Future studies are needed to investigate the relevant dynamics (i.e., wind-forced sea-ice deformation and breakup) and thermodynamics (e.g., ocean-to-ice heat fluxes) of the sea-ice pack to explore the precise mechanisms by which the sea-ice concentration relates to the structure of the FSD, and how this relationship might differ in different settings. For example, a scenario might be envisioned where the FSD slope could steepen (e.g., as a result of fewer large floes, and more smaller floes) while the SIC remains the same, and this might indicate a fracturing. Conversely, a shoaling of the FSD slope associated with the loss of small floes (e.g., via relatively rapid lateral melt of smaller floes compared to larger floes) may be associated with a different SIC-FSD relationship.

Finally, we point out that several other previous studies report two distinct floe-size regimes, in which a small-floe regime is characterized by shallower FSD slopes and a large-floe regime by steeper slopes (Geise et al., 2017; Steer et al., 2008; Toyota et al., 2011; Toyota et al., 2006). Using images from the Weddell Sea, Steer et al. (2008) examine the non-cumulative FSD for floe diameters between O(1) and O(100) m, finding a change in FSD slope at 20 m. In addition, Perovich and Jones (2014) show a possible plateauing of the FSD slope at the small-floe scale (although they do not explicitly refer to two regimes). For a range of sea-ice settings, and considering floe diameters in the range O(1-1,000) m, Toyota et al. (2011) and Toyota et al. (2006) find two floe-size regimes for floe sizes larger and smaller than about 20 to 40 m diameter. We note that these studies classify images into ice and water as an initial step, choosing a classification threshold. Our test of FSD sensitivity to this choice reveals that the FSD can appear divided into two power-law regimes if this choice does not adequately identify small floes.

Our finding of a single power-law suggests that the processes which govern the distribution of floe sizes are similar across the full range of floe sizes, while studies which find two distinct power-law regimes would indicate that different processes act on different scales. For example, Horvat and Tziperman (2017) use a coupled ice-ocean model to show that increased lateral melt on specific floe sizes and transient oceanic forcing on the ice pack can perturb the FSD behavior from a single power-law at the relevant scale. Future work is needed to determine how different FSD structures might emerge in certain settings.

**Appendix A**

| Image GFL site | Date | Lat (DD) Lon (DD) | Area (km$^2$) | Tot. Ice Area (km$^2$) | SIC (CDR) | Dist. to Ice Edge (km) | Mean Daily SAT (2-m, °C) | Analysis Results | | | | | | | |
|---|---|---|---|---|---|---|---|---|---|---|---|---|---|---|---|
| | | | | | | | | Tot. # Whole Floes | Tot. Whole-Floe Area (km$^2$) | # Floes in Slope-Fitting Range | $m \pm e$ | $m_{MLE} \pm e_{MLE}$ | $a_{min}$ (MLE, m$^2$) | $p$-value (MLE) | Seg. Conf. |
| **1** B | 28 Jul 1999 | 73.0 -149.9 | 258 | 192 | 0.74 (0.87) | 500 | 0.05 | 898,684 | 138 | 173,089 | -1.95 ±0.04 | -2.00 ±0.00 | 164 | 0.65 | L |

| | Date | Lat/Lon | | | | | | | | | | | | | |
|---|---|---|---|---|---|---|---|---|---|---|---|---|---|---|---|
| **2** NCB | 22 May 2000 | 85.0 -120.0 | 213 | 201 | 0.94 (1.00) | 2,400 | -8.35 | 493,151 | 119 | 45,316 | -1.77 ±0.05 | -1.71 ±0.01 | 558 | 0.19 | H |
| **3** NCB | 27 Jul 2000 | 85.1 -119.4 | 50 | 41 | 0.82 (1.00) | 1,900 | -0.04 | 91,756 | 30 | 13,062 | -1.75 ±0.07 | -1.70 ±0.01 | 345 | 0.93 | L |
| **4** NCB | 27 Jul 2000 | 85.1 -120.8 | 51 | 44 | 0.85 (1.00) | 1,800 | -0.03 | 97,913 | 20 | 14,355 | -1.78 ±0.03 | -1.77 ±0.01 | 151 | 0.47 | L |
| **5** NCB | 15 Aug 2000 | 84.9 -118.8 | 88 | 76 | 0.86 (0.94) | 1,600 | -0.42 | 93,122 | 51 | 10,489 | -1.65 ±0.04 | -1.59 ±0.01 | 443 | 0.69 | H |
| **6** NCB | 15 Aug 2000 | 85.0 -119.6 | 84 | 72 | 0.86 (0.93) | 1,600 | -0.45 | 227,144 | 51 | 38,658 | -1.82 ±0.07 | -1.80 ±0.01 | 131 | 0.01 | H |
| **7** B | 26 Aug 2000 | 72.9 -149.7 | 100 | 11 | 0.12 (0.31) | 200 | -0.22 | 84,795 | 9 | 15,450 | -2.00 ±0.10 | -1.94 ±0.01 | 178 | 1.00 | L |
| **8** NCB | 29 Aug 2000 | 85.0 -118.9 | 282 | 203 | 0.72 (1.00) | 1,600 | -1.23 | 196,189 | 121 | 20,857 | -1.71 ±0.09 | -1.59 ±0.03 | 2,536 | 0.87 | H |
| **9** B | 2 Sep 2000 | 73.0 -150.2 | 161 | 42 | 0.26 (0.31) | 0 | 1.49 | 343,582 | 41 | 62,044 | -1.99 ±0.05 | -1.88 ±0.00 | 12 | 0.00 | L |
| **10** NCB | 2 Sep 2000 | 85.1 -119.9 | 103 | 93 | 0.91 (1.00) | 1,400 | -2.14 | 96,260 | 32 | 10,253 | -1.66 ±0.08 | -1.59 ±0.02 | 996 | 0.35 | H |
| **11** B | 21 May 2001 | 73.0 -149.7 | 208 | 200 | 0.96 (1.00) | 1,400 | -6.09 | 120,082 | 45 | 13,167 | -1.68 ±0.08 | -1.72 ±0.02 | 526 | 0.39 | H |
| **12** B | 16 May 2002 | 73.0 -150.0 | 324 | 305 | 0.94 (1.00) | 1,200 | -7.55 | 101,960 | 71 | 34,096 | -1.77 ±0.05 | -1.70 ±0.02 | 1,785 | 0.92 | H |
| **13** NCB | 21 May 2002 | 85.0 -120.0 | 329 | 312 | 0.95 (1.00) | 2,500 | -7.67 | 254,595 | 81 | 28,514 | -1.82 ±0.07 | -1.69 ±0.02 | 1,225 | 0.87 | H |
| **14** B | 23 May 2002 | 73.0 -149.9 | 139 | 133 | 0.96 (0.99) | 1,200 | -1.62 | 32,975 | 16 | 9,325 | -1.74 ±0.05 | -1.71 ±0.02 | 287 | 0.65 | H |
| **15** B | 23 May 2002 | 73.0 -150.1 | 135 | 127 | 0.94 (0.99) | 1,200 | -1.62 | 42,216 | 23 | 6,999 | -1.69 ±0.03 | -1.72 ±0.01 | 111 | 0.78 | H |
| **16** B | 13 May 2006 | 73.0 -150.0 | 338 | 288 | 0.85 (1.00) | 1,400 | -1.66 | 384,971 | 169 | 45,665 | -1.72 ±0.02 | -1.69 ±0.01 | 686 | 0.81 | L |
| **17** C | 12 Jun 2006 | 70.0 -170.0 | 217 | 135 | 0.62 (0.85) | 100 | 0.29 | 1,478,840 | 124 | 185,970 | -1.95 ±0.02 | -1.92 ±0.01 | 938 | 0.99 | L |
| **18** NCB | 23 Jul 2007 | 85.0 -119.9 | 266 | 212 | 0.80 (1.00) | 1,600 | 0.29 | 616,906 | 179 | 94,747 | -1.80 ±0.04 | -1.75 ±0.00 | 104 | 0.73 | L |
| **19** C | 12 Jun 2008 | 70.0 -170.0 | 307 | 271 | 0.88 (1.00) | 200 | 0.35 | 287,682 | 136 | 48,579 | -1.72 ±0.04 | -1.66 ±0.01 | 546 | 0.12 | H |
| **20** B | 8 Apr 2010 | 73.0 -150.0 | 182 | 174 | 0.95 (1.00) | 1,900 | -18.03 | 119,100 | 27 | 11,590 | -1.73 ±0.07 | -1.69 ±0.02 | 411 | 0.83 | H |
| **21** B | 29 Apr 2011 | 73.0 -150.0 | 226 | 214 | 0.95 (1.00) | 1,800 | -12.25 | 52,232 | 10 | 9,135 | -1.77 ±0.11 | -1.81 ±0.03 | 322 | 0.43 | H |
| **22** NCB | 29 Apr 2011 | 85.0 -120.0 | 368 | 334 | 0.91 (1.00) | 3,100 | -13.51 | 273,344 | 83 | 43,775 | -1.78 ±0.06 | -1.71 ±0.01 | 351 | 0.55 | H |
| **23** B | 29 May 2011 | 73.0 -150.0 | 230 | 215 | 0.94 (1.00) | 1,100 | -1.70 | 198,991 | 59 | 21,181 | -1.85 ±0.04 | -1.82 ±0.01 | 124 | 0.71 | L |
| **24** B | 23 May 2012 | 73.0 -149.9 | 98 | 86 | 0.88 (0.94) | 1,200 | -0.88 | 32,335 | 15 | 7,314 | -1.72 ±0.07 | -1.65 ±0.02 | 513 | 0.67 | H |
| **25** C | 5 Jul 2012 | 70.0 -170.0 | 123 | 39 | 0.31 (0.54) | 0 | 1.22 | 154,212 | 28 | 28,092 | -1.88 ±0.09 | -1.82 ±0.00 | 32 | 0.07 | H |
| **26** IB | 19 Apr 2013 | 81.1 -110.3 | 47 | 44 | 0.93 (1.00) | 3,300 | -15.14 | 38,461 | 6 | 6,218 | -1.87 ±0.10 | -1.86 ±0.01 | 56 | 0.76 | H |

| | | | | | | | | | | | | | | | |
|---|---|---|---|---|---|---|---|---|---|---|---|---|---|---|---|
| **27** IB | 19 Apr 2013 | 81.1 -121.0 | 53 | 51 | 0.97 (1.00) | 3,100 | -14.08 | 13,598 | 2 | 2,758 | -1.81 ±0.06 | -1.76 ±0.03 | 140 | 0.70 | H |
| **28** IB | 20 Apr 2013 | 78.0 -126.0 | 67 | 64 | 0.96 (1.00) | 2,900 | -11.04 | 31,123 | 10 | 9,046 | -1.83 ±0.08 | -1.76 ±0.02 | 192 | 0.20 | L |
| **29** IB | 20 Apr 2013 | 77.4 -121.1 | 63 | 55 | 0.88 (1.00) | 3,000 | -11.06 | 97,533 | 35 | 32,464 | -1.85 ±0.07 | -1.76 ±0.01 | 575 | 0.54 | L |
| **30** IB | 22 Apr 2013 | 82.0 -95.0 | 97 | 89 | 0.91 (1.00) | 3,600 | -17.67 | 98,873 | 22 | 22,487 | -1.82 ±0.07 | -1.85 ±0.01 | 247 | 0.02 | L |
| **31** IB | 22 Apr 2013 | 82.8 -106.2 | 97 | 81 | 0.83 (1.00) | 3,400 | -17.76 | 162,725 | 39 | 27,188 | -1.84 ±0.08 | -1.84 ±0.01 | 53 | 0.00 | L |
| **32** IB | 22 Apr 2013 | 82.2 -153.0 | 102 | 97 | 0.95 (1.00) | 2,900 | -17.97 | 41,172 | 26 | 14,095 | -1.76 ±0.08 | -1.69 ±0.02 | 730 | 0.82 | H |
| **33** IB | 22 Apr 2013 | 82.0 -140.8 | 102 | 89 | 0.88 (1.00) | 2,900 | -17.71 | 62,951 | 36 | 13,951 | -1.74 ±0.09 | -1.66 ±0.01 | 494 | 0.50 | H |
| **34** IB | 22 Apr 2013 | 80.0 -114.0 | 100 | 92 | 0.92 (1.00) | 3,200 | -15.28 | 93,149 | 26 | 19,682 | -1.78 ±0.07 | -1.81 ±0.01 | 271 | 0.93 | L |
| **35** B | 24 Apr 2013 | 73.0 -150.1 | 107 | 102 | 0.95 (1.00) | 2,000 | -10.57 | 37,547 | 23 | 12,255 | -1.77 ±0.03 | -1.72 ±0.02 | 458 | 0.99 | H |
| **36** C | 27 Apr 2013 | 70.0 -170.0 | 214 | 177 | 0.83 (0.95) | 1,400 | -7.02 | 161,051 | 117 | 26,088 | -1.67 ±0.03 | -1.67 ±0.01 | 536 | 0.91 | H |
| **37** NCB | 6 May 2013 | 85.0 -120.0 | 99 | 94 | 0.95 (1.00) | 2,400 | -13.60 | 75,432 | 23 | 15,516 | -1.78 ±0.06 | -1.85 ±0.02 | 194 | 0.03 | H |
| **38** C | 9 May 2013 | 70.0 -170.0 | 242 | 202 | 0.83 (1.00) | 500 | -0.72 | 199,112 | 87 | 27,434 | -1.71 ±0.04 | -1.80 ±0.01 | 49 | 0.00 | H |
| **39** NCB | 20 May 2013 | 85.0 -119.6 | 85 | 80 | 0.94 (1.00) | 2,400 | -9.51 | 65,391 | 12 | 13,755 | -1.88 ±0.06 | -1.85 ±0.01 | 146 | 0.78 | H |
| **40** NCB | 20 May 2013 | 85.0 -120.1 | 86 | 80 | 0.93 (1.00) | 2,400 | -9.49 | 51,493 | 17 | 6,800 | -1.72 ±0.07 | -2.01 ±0.01 | 11 | 0.00 | H |
| **41** NCB | 20 May 2013 | 85.0 -120.7 | 86 | 80 | 0.94 (1.00) | 2,400 | -9.47 | 38,773 | 19 | 7,811 | -1.70 ±0.08 | -2.01 ±0.01 | 20 | 0.00 | H |
| **42** C | 31 May 2013 | 69.9 -170.0 | 83 | 73 | 0.89 (1.00) | 500 | 0.32 | 34,561 | 16 | 11,386 | -1.74 ±0.08 | -1.68 ±0.02 | 683 | 0.24 | H |
| **43** C | 10 Jun 2013 | 70.0 -170.1 | 135 | 97 | 0.72 (1.00) | 200 | 1.63 | 165,353 | 81 | 49,902 | -1.84 ±0.09 | -1.71 ±0.01 | 376 | 0.04 | H |
| **44** C | 10 Jun 2013 | 70.0 -169.9 | 69 | 45 | 0.65 (1.00) | 200 | 1.59 | 91,001 | 31 | 19,980 | -1.75 ±0.04 | -1.71 ±0.01 | 308 | 0.05 | H |
| **45** B | 12 Jun 2013 | 73.0 -150.0 | 251 | 200 | 0.80 (0.98) | 800 | 1.21 | 285,207 | 150 | 93,676 | -1.80 ±0.04 | -1.74 ±0.01 | 314 | 0.20 | H |
| **46** B | 12 Jun 2013 | 73.0 -150.0 | 252 | 165 | 0.66 (0.98) | 800 | 1.21 | 312,992 | 121 | 74,241 | -1.82 ±0.03 | -1.80 ±0.01 | 335 | 0.28 | H |
| **47** SIZRS | 21 Jun 2013 | 71.0 -150.0 | 177 | 121 | 0.68 (0.92) | 700 | 2.22 | 213,969 | 87 | 60,905 | -1.84 ±0.02 | -1.90 ±0.01 | 159 | 0.00 | H |
| **48** NCB | 26 Jun 2013 | 85.0 -120.0 | 355 | 324 | 0.91 (1.00) | 2,100 | 0.42 | 220,637 | 221 | 65,672 | -1.71 ±0.03 | -1.67 ±0.01 | 637 | 0.53 | H |
| **49** C | 27 Jun 2013 | 70.0 -170.0 | 236 | 99 | 0.42 (0.53) | 200 | 3.68 | 871,075 | 95 | 135,222 | -2.03 ±0.07 | -1.89 ±0.00 | 30 | 0.01 | L |
| **50** SIZRS | 14 Jul 2013 | 71.0 -150.0 | 572 | 158 | 0.28 (0.48) | 200 | -0.60 | 550,413 | 117 | 166,449 | -1.96 ±0.03 | -1.94 ±0.01 | 1,289 | 0.99 | H |
| **51** IB | 8 Apr 2014 | 81.4 -128.4 | 101 | 95 | 0.94 (1.00) | 2,500 | -25.42 | 42,772 | 12 | 14,465 | -1.87 ±0.08 | -1.82 ±0.03 | 655 | 0.96 | H |

| # | Date | Lat/Lon | | | | | | | | | | | | | |
|---|---|---|---|---|---|---|---|---|---|---|---|---|---|---|---|
| 52 C | 17 Apr 2014 | 70.0 -170.0 | 205 | 167 | 0.82 (0.93) | 1,000 | -11.18 | 295,680 | 50 | 37,869 | -1.81 ±0.05 | -1.75 ±0.01 | 642 | 0.42 | H |
| 53 MIZ | 24 Apr 2014 | 72.5 -138.0 | 902 | 885 | 0.98 (0.98) | 1,700 | -8.33 | 49,453 | 44 | 27,301 | -1.77 ±0.08 | -1.68 ±0.03 | 4,151 | 1.00 | H |
| 54 MIZ | 24 Apr 2014 | 73.4 -137.2 | 1,014 | 963 | 0.95 (0.99) | 1,800 | -9.53 | 172,863 | 118 | 62,479 | -1.81 ±0.07 | -1.79 ±0.01 | 902 | 0.58 | H |
| 55 MIZ | 24 Apr 2014 | 74.2 -136.2 | 978 | 942 | 0.96 (1.00) | 1,900 | -11.18 | 87,901 | 146 | 42,755 | -1.86 ±0.05 | -1.78 ±0.02 | 1,702 | 0.68 | H |
| 56 SIZRS | 25 Apr 2014 | 79.0 -150.0 | 378 | 366 | 0.97 (1.00) | 2,100 | -10.73 | 57,214 | 80 | 19,730 | -1.71 ±0.04 | -1.69 ±0.02 | 2,092 | 0.99 | H |
| 57 B | 28 Apr 2014 | 73.0 -150.0 | 8 | 8 | 0.91 (1.00) | 1,500 | -12.57 | 13,689 | 1 | 1,870 | -1.83 ±0.06 | -2.02 ±0.01 | 12 | 0.00 | L |
| 58 B | 28 Apr 2014 | 73.0 -150.0 | 237 | 229 | 0.97 (1.00) | 1,500 | -12.57 | 49,303 | 29 | 10,255 | -1.71 ±0.07 | -1.70 ±0.02 | 370 | 0.97 | H |
| 59 MIZ | 30 Apr 2014 | 73.0 -141.0 | 1,047 | 933 | 0.89 (0.98) | 1,700 | -12.52 | 120,076 | 123 | 36,373 | -1.73 ±0.05 | -1.64 ±0.02 | 2,394 | 1.00 | H |
| 60 MIZ | 30 Apr 2014 | 73.7 -140.2 | 907 | 826 | 0.91 (0.99) | 1,700 | -13.24 | 171,852 | 112 | 55,788 | -1.78 ±0.03 | -1.77 ±0.01 | 329 | 0.84 | H |
| 61 MIZ | 30 Apr 2014 | 74.5 -139.3 | 818 | 751 | 0.92 (1.00) | 1,800 | -13.54 | 221,074 | 290 | 66,748 | -1.74 ±0.05 | -1.73 ±0.01 | 430 | 0.86 | H |
| 62 SIZRS | 2 May 2014 | 71.0 -150.0 | 307 | 269 | 0.87 (1.00) | 800 | 0.28 | 148,708 | 22 | 34,978 | -1.91 ±0.08 | -1.93 ±0.02 | 429 | 0.50 | H |
| 63 NCB | 21 May 2014 | 85.0 -120.0 | 222 | 215 | 0.96 (1.00) | 2,100 | -5.62 | 125,882 | 37 | 28,435 | -1.80 ±0.05 | -1.77 ±0.01 | 336 | 0.64 | H |
| 64 MIZ | 27 May 2014 | 73.2 -138.3 | 261 | 247 | 0.94 (1.00) | 1,200 | -5.56 | 76,033 | 59 | 30,349 | -1.75 ±0.04 | -1.68 ±0.01 | 565 | 0.38 | H |
| 65 MIZ | 30 May 2014 | 73.1 -138.7 | 1,153 | 1,049 | 0.91 (1.00) | 1,200 | -4.39 | 247,707 | 641 | 61,214 | -1.67 ±0.01 | -1.68 ±0.01 | 105 | 0.46 | H |
| 66 SIZRS | 30 May 2014 | 76.0 -150.0 | 457 | 407 | 0.89 (1.00) | 1,100 | -4.90 | 140,703 | 168 | 59,351 | -1.70 ±0.05 | -1.65 ±0.01 | 602 | 0.09 | H |
| 67 SIZRS | 13 Jun 2014 | 80.0 -150.0 | 594 | 536 | 0.90 (1.00) | 1,200 | 0.51 | 319,004 | 454 | 120,494 | -1.69 ±0.04 | -1.65 ±0.00 | 196 | 0.00 | H |
| 68 MIZ | 17 Jun 2014 | 73.2 -146.5 | 619 | 571 | 0.92 (1.00) | 600 | 1.24 | 169,888 | 147 | 72,240 | -1.82 ±0.05 | -1.81 ±0.01 | 1,790 | 1.00 | H |
| 69 SIZRS | 17 Jun 2014 | 75.0 -149.8 | 433 | 384 | 0.89 (1.00) | 700 | 0.43 | 160,564 | 113 | 74,497 | -1.83 ±0.07 | -1.75 ±0.02 | 2,489 | 0.93 | H |
| 70 MIZ | 19 Jun 2014 | 73.5 -141.4 | 761 | 585 | 0.77 (0.96) | 800 | 0.70 | 1,021,902 | 398 | 261,365 | -1.85 ±0.03 | -1.95 ±0.01 | 34 | 0.00 | H |
| 71 SIZRS | 20 Jun 2014 | 80.0 -150.0 | 384 | 337 | 0.88 (1.00) | 1,200 | 0.85 | 204,119 | 294 | 67,375 | -1.68 ±0.04 | -1.65 ±0.00 | 223 | 0.00 | H |
| 72 SIZRS | 21 Jun 2014 | 79.0 -150.1 | 342 | 311 | 0.91 (1.00) | 1,100 | 0.87 | 188,082 | 262 | 62,817 | -1.69 ±0.03 | -1.66 ±0.01 | 434 | 0.33 | H |
| 73 MIZ | 11 Jul 2014 | 74.4 -142.3 | 865 | 696 | 0.80 (1.00) | 300 | 0.91 | 493,562 | 574 | 133,660 | -1.70 ±0.03 | -1.69 ±0.00 | 77 | 0.13 | H |
| 74 MIZ | 31 Jul 2014 | 74.6 -140.1 | 659 | 467 | 0.71 (1.00) | 300 | 0.21 | 281,260 | 399 | 92,856 | -1.74 ±0.05 | -1.66 ±0.00 | 37 | 0.00 | L |
| 75 MIZ | 11 Aug 2014 | 73.6 -156.0 | 585 | 259 | 0.44 (0.64) | 100 | -0.54 | 340,880 | 201 | 103,820 | -1.82 ±0.05 | -1.89 ±0.01 | 2,385 | 0.65 | L |
| 76 MIZ | 14 Aug 2014 | 73.6 -157.2 | 857 | 303 | 0.35 (0.66) | 0 | -0.37 | 495,291 | 277 | 177,019 | -1.85 ±0.04 | -1.91 ±0.01 | 2,385 | 0.89 | L |

| | | | Image area | Total ice area | SIC (CDR) | Dist. to edge | SAT | No. whole floes | Whole-floe area | No. floes fit range | $m \pm e$ | $m_{MLE} \pm e_{MLE}$ | $a_{min}$ | $p$ | Conf. |
|---|---|---|---|---|---|---|---|---|---|---|---|---|---|---|---|
| **77** MIZ | 20 Sep 2014 | 77.3 -139.3 | 846 | 740 | 0.87 (1.00) | 400 | -9.63 | 494,485 | 556 | 186,074 | -1.80 ±0.05 | -1.92 ±0.02 | 9,808 | 0.68 | H |
| **78** MIZ | 26 Sep 2014 | 77.2 -140.4 | 366 | 328 | 0.90 (1.00) | 400 | -9.87 | 71,060 | 30 | 25,990 | -1.90 ±0.08 | -1.90 ±0.02 | 352 | 1.00 | H |

**Table A1. Image number with corresponding GFL site (where B = Beaufort Sea, C = Chukchi Sea, and NCB = Northern Canada Basin are the fixed sites; and IB = IceBridge, SIZRS = Seasonal Ice Zone Reconnaissance Surveys, and MIZ = Marginal Ice Zone are corresponding program locations), date, latitude and longitude (decimal degrees), image area (total area viewed by the sensor, rounded to the nearest km², total ice area (combined area of all ice identified in the image including that of border-intersecting floes, rounded to the nearest km²), sea ice concentration (fractional area, SIC, where the first value is derived from the image and the second, in parentheses, from the NOAA/NSIDC Climate Data Record, CDR), approximate distance to the median ice edge of that month (rounded to the nearest hundred km), SAT (2-m mean daily, °C), total number of whole floes retrieved (between 5 m² and 100 km² with the requirement of 2 per bin), total whole-floe area (combined area of all whole floes identified in the image after clearing image border-intersecting floes, rounded to the nearest km²), number of floes with areas in the slope-fitting range (between 50 m² and 5 km²), FSD linear best-fit slope $m \pm$ slope fit error $e$ (95% confidence), FSD maximum likelihood estimate slope $m_{MLE} \pm$ slope fit uncertainty $e_{MLE}$ (standard deviation over iterative fits), strict lower bound on MLE-fitted power-law behavior $a_{min}$, $p$-value on the MLE-fitted power-law model, and segmentation confidence (where L = Low and H = High). Rows in light gray indicate images acquired on the same day at the same site. SIC is calculated by dividing the total ice area by the image area; note that this calculation is performed prior to rounding the areas displayed in this table. SIC CDR is from the NOAA/NSIDC Climate Data Record of Passive Microwave Sea Ice Concentration, Version 4 (Peng et al., 2013; Meier et al., 2021). Median ice edge contours are from the NSIDC Sea Ice Index, Version 3 (Fetterer et al., 2017). SAT data are from the ECMWF ERA5 Reanalysis hourly data on single levels from 1979 to present (Hersbach et al., 2018).**

## Code and Data Availability

The MATLAB algorithm (Denton, 2022) written and used here to segment sea ice floes in satellite images is available at https://doi.org/10.5281/zenodo.6146144. The sea-ice floe segmentation products (Denton and Timmermans, 2022) derived from MEDEA imagery and presented here are available for download at https://doi.org/10.5281/zenodo.6341621. MEDEA images are available from the USGS GFL (https://www.usgs.gov/core-science-systems/nli/global-fiducials-library). SIC passive microwave data are from the NOAA/NSIDC Climate Data Record of Passive Microwave Sea Ice Concentration, Version 4 (Peng et al., 2013; Meier et al., 2021). Median ice edge contours are from the NSIDC Sea Ice Index, Version 3 (Fetterer et al., 2017). SAT data are from the ECMWF ERA5 Reanalysis (Hersbach et al., 2020), ERA5 hourly data on single levels from 1979 to present (Hersbach et al., 2018), and were downloaded from the Copernicus Climate Change Service (C3S) Climate Data Store. The results contain modified Copernicus Climate Change Service information 2021. Neither the European Commission nor ECMWF is responsible for any use that may be made of the Copernicus information or data it contains.

## Author Contribution

A.D. formulated the image segmentation algorithm, performed the analysis, and took the lead in writing the manuscript. All authors shaped the research, analysis, and writing.

## Competing Interests

The authors declare that they have no conflict of interest.

## Acknowledgments

This research was funded by the Office of Naval Research Multidisciplinary University Research Initiative (MURI) on Mathematics and Data Science for Physical Modeling and Prediction of Sea Ice.

The authors would like to thank Hans C. Graber of The University of Miami Center for Southeastern Tropical Advanced Remote Sensing for providing several of the MEDEA images used in this study.

Values $m_{MLE}$, $e_{MLE}$, $a_{min}$, and the corresponding $p$-values were computed directly using Aaron Clauset's MATLAB power-law fitting package, available at https://aaronclauset.github.io/powerlaws/). We thank the Editor, Chris Horvat, and one anonymous reviewer for their valuable comments and suggestions.

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
