# Peer review of "Characterizing the Sea-Ice Floe Size Distribution in the Canada Basin from High-Resolution Optical Satellite Imagery"

_The Cryosphere, 2021_

## Referee Comment (RC1)

Review for *The Cryosphere* of

Characterizing the Sea-Ice Floe Size Distribution in the Canada Basin from High-Resolution Optical Satellite Imagery

by Denton and Timmermans

The authors analyzed 78 images from the Canada Basin of the Arctic Ocean. The images have 1-meter spatial resolution and span the period 1999-2014 during the months April to September. They identified individual ice floes in the images and analyzed the floe size distribution (FSD), where size is measured by floe area. They have four main results: (1) The FSDs follow a power-law distribution between areas $5 \times 10^1$ m$^2$ and $5 \times 10^6$ m$^2$ (50 m$^2$ to 5 km$^2$) with power-law exponents ranging from $-1.65$ to $-2.03$. (2) The FSDs are sensitive to the threshold used to separate ice from water in the images. Other studies may have erroneously found two power-law regimes by not setting the proper threshold between ice and water. (3) A linear relationship is found between power-law exponents and sea-ice concentration (SIC), with more negative exponents corresponding to lower SIC. (4) Locations that experience a seasonal cycle in SIC also have a seasonal variation in power-law exponent, but sites with high year-round SIC do not.

The analysis and conclusions of this paper are generally sound. I recommend publication after the authors consider the following comments and suggestions, which are given in page order.

Comments and suggestions

Lines 9-11. "the structure of the FSD is found to be sensitive to a classification threshold value… and an objective approach to minimize this sensitivity is presented." I searched throughout the paper for the objective approach, but all I could find was this (on lines 163-164): "we iteratively increase the threshold above the minimum until the edges of small floes are appropriately delineated." I have no objection to this method, but I wouldn't call it an objective approach that minimizes anything. It sounds like "visual inspection" or "manual selection" to me. If that's what it is, please say so. If I'm missing the objective approach, please provide more detail.

Lines 182-189. This describes the construction of the FSD by binning the data and fitting a line (in log-log space) to the binned values. That's fine, but it's a shame that the authors did not use Maximum Likelihood Estimation to find the best-fitting exponent, which does not require binning the data (and hence avoids the problem of having an adequate number of samples per bin), and which gives a more accurate estimate of the exponent than least-squares fitting of binned data. No changes necessary, I just wanted to bring up this point.

Lines 199-203. The authors choose to use floe area as the measure of floe size, because once the pixels of a floe have been identified, it's a simple matter to count them and multiply by the area per pixel. However, it's also a simple matter to directly calculate the mean caliper diameter (MCD) from the coordinates of the floe pixels. I'm not talking about an approximation that relates the average MCD to the average area (as in Rothrock and Thorndike, 1984) – I'm saying that the MCD can be easily calculated exactly for every individual floe, as follows.

In a pixel-based coordinate system with (0,0) at the lower left corner of the image, let $(x_k, y_k)$ be the pixel coordinates that comprise a floe, denoted by vectors **x** and **y**. Then:

```
for angle = 0, 179 do begin          ; Loop over caliper angles from 0° to 179° in 1° increments
   c = cos(angle * pi / 180.)
   s = sin(angle * pi / 180.)
   r = c*x + s*y
   diameter(angle) = max(r) – min(r) ; This is the caliper diameter at the given angle
endfor
MCD = total(diameter)/180.           ; Average over all angles of the calipers to get MCD
```

There's nothing wrong with using floe area as the measure of floe size, but in 18 previous studies of the FSD, 17 of them have used a diameter or dimensional length, and only one has used area. It would be easier to compare the results of this study with previous results if the authors had used MCD or another length scale. No changes necessary, I just wanted to bring up this point.

Lines 216-218. "the slope of the FSD is valid only for floe areas larger than 50 $m^2$ and we limit fitting in log-log plots (to estimate *m*) to floe areas between 50 $m^2$ and 5 $km^2$"
How many floes fall outside these bounds, roughly? Is it 10% of all floes? 50%?
Table A1 gives the "number of floes (whole) retrieved" (according to the caption on page 16).
Does the number in the table include only the floes between the given bounds, or all floes?
If it's all floes, then another column should be included in the table giving the number of floes with area between the bounds.

Line 240 and Figure 3 caption. Some of the images are segmented (divided into ice and water) with high confidence and others with low confidence. Table A1 should indicate which images are segmented with high confidence and which with low confidence.

Figure 3 panels (b) and (c) include "error bars representing one standard deviation" (lines 252 and 254). However, in some cases the number of samples is 1, so it makes no sense to calculate a standard deviation (which is 0) and call it an error bar. The number of samples for a given month and location never exceeds 7, and in most cases it's 1, 2, or 3. It's of dubious statistical rigor and utility to calculate standard deviations and call them error bars in these cases. Furthermore, the error bars make the plots more difficult to read. In my opinion, all the error bars should be removed from panels (b) and (c), and reference to them deleted from the caption.

Figure 3 panels (b) and (c) also include "mean monthly slopes" and "mean monthly SICs."
In my comment below about Table A1, I note that not all the images provide independent estimates of sea-ice properties because some images were acquired on the same day of the same year at essentially the same location. For example, images #39, #40, and #41 are all from 20 May 2013 at the Northern Canada Basin site (see Table A1). Therefore, in calculating mean monthly slopes for May, the values from #39, #40, and #41 should be averaged first, and then their average should be averaged with the other May values from the same site (i.e., images #2, #13, #37, and #63). Perhaps the authors have actually calculated the means in this way. If not, I'm sure it wouldn't make much difference in the final results, but I just wanted to bring up the point about statistical non-independence of the images.

Lines 290-299. This paragraph is about the lack of a relationship between surface air temperature (SAT) and power-law exponent $m$ of the FSD. The lack of a relationship is not surprising. During the Arctic summer, the SAT over ice is pegged at the melting point of ice because the surface is an ice bath (ice and water). For example, see the plots of SAT in Rigor et al. (2000). From the time of melt onset to the time of freeze-up (roughly June through August in the Beaufort Sea) there is essentially no variability in the SAT over ice, so it can't possibly explain the variability in FSD. The solar energy that goes into the ocean melts ice, potentially changing the FSD, but it does not raise the SAT over ice. In my opinion, it would have made more sense to look for a connection between $m$ (FSD) and some other thermodynamic variable that characterizes ocean heat content or energy balance.

Rigor, I.G., R.L. Colony, and S. Martin (2000). Variations in Surface Air Temperature Observations in the Arctic, 1979–97. *J. Climate*, pp. 896-914, https://doi.org/10.1175/1520-0442(2000)013<0896:VISATO>2.0.CO;2

Lines 303-307. The authors state that a power-law probability density function (pdf) of the form $n(a) \sim a^m$ where $a$ is floe area is equivalent to a power-law pdf of the form $x^{2m+1}$ where $x$ is floe diameter. This is correct, but the reason given is wrong. It has nothing to do with the binning of data or the normalization of bin counts. It follows from a straightforward application of basic probability theory – see the notes at the end of this review. Lines 303-307 should be re-written to simply state that $a^m$ is equivalent to $x^{2m+1}$ as a result of basic probability theory.

Line 315. "image not included in our analysis due to partial cloud-cover"
Somewhere in the paper the authors should state how many images were rejected due to partial or total cloud cover, and how those decisions were made. Was it by visual inspection?

Lines 346-350. "seasonal variation in $m$ is more directly related to changes in SIC" (than SAT).
Yes, but as noted above, there's no reason to expect a connection between $m$ and SAT.
"Future studies are needed to investigate the relevant dynamics … and thermodynamics…"
Yes, undoubtedly the same forces that drive changes in SIC also drive changes in FSD. This raises the question: why try to relate SIC and FSD in the first place? Neither one drives the other, they're both the result of underlying dynamic and thermodynamic forcing. It seems like SIC could never be more than an imperfect reflection of FSD. Imagine an image with a power-law FSD and a certain value of SIC. Now double the size of the image by adding only ocean pixels. The FSD remains exactly the same, but the SIC is cut in half. Conversely, it's easy to imagine a scenario in which the power-law exponent of the FSD changes but the SIC does not. What is the motivation for relating FSD to SIC? Wouldn't it make more sense to look for connections between FSD and, say, wind stress?

Table A1. This is a good and useful table, but it is incomplete. As noted above, it should include the number of floes with area between 50 m$^2$ and 5 km$^2$ (i.e., the bounds used in determining the power-law exponent $m$) as well as the total number of floes (which is given already, I believe).
    Also, the table should indicate which images were segmented with high confidence and which with low confidence.
    Also, there should be a column to indicate whether an image is from the Beaufort fiducial site, the Chukchi fiducial site, the Northern Canada Basin fiducial site, or another site. I realize

that this can be inferred from the latitude and longitude, but it is extremely tedious to go through the table and extract that information.

Another point that should be noted somewhere in the paper is that some of the images were acquired on the same day of the same year at essentially the same location, and therefore do not provide independent estimates of sea-ice properties (FSD and SIC). In particular:

**3 and #4 are both 27 July 2000 at the N. Basin site**
**5 and #6 are both 15 Aug 2000 at the N. Basin site**
**14 and #15 are both 23 May 2002 at the Beaufort site**
**39 and #40 and #41 are all 20 May 2013 at the N. Basin site**
**43 and #44 are both 10 June 2013 at the Chukchi site**
**45 and #46 are both 12 June 2013 at the Beaufort site**
**57 and #58 are both 28 April 2014 at the Beaufort site**

I found it useful to summarize the number of images by location and month in a table. Consider including something like this in the paper:

|       | Beaufort | Chukchi | N. Basin | Other | Total |
|-------|----------|---------|----------|-------|-------|
| **April** | 5 (*) | 2 | 1 | 17 | 25 |
| **May** | 7 (*) | 2 | 7 (*) | 4 | 20 |
| **June** | 2 (*) | 5 (*) | 1 | 7 | 15 |
| **July** | 1 | 1 | 3 (*) | 3 | 8 |
| **Aug.** | 1 | 0 | 3 (*) | 2 | 6 |
| **Sept.** | 1 | 0 | 1 | 2 | 4 |
| **Total** | 17 | 10 | 16 | 35 | 78 |

The (*) symbol indicates that some of the images are from the same day of the same year at essentially the same location, i.e., not independent samples.
**Beaufort** site is shown in Figure 1(a) at the letter e.
**Chukchi** site is shown in Figure 1(a) at the letter c.
**N. Basin** site is shown in Figure 1(a) at the letter d.
**Other** sites are shown in Figure 1(a) at white dots.

Minor typographical notes

Line 6. Change "atmosphere and ocean" to "atmospheric and oceanic" as on line 22.

Line 64. "Aeronautical" should be "Aeronautics"

Line 66. "The images AT FIDUCIAL SITES are panchromatic…"

Last sentence on page 3. Please say that the surface air temperature (SAT) is at the 2-meter level.

Figure 1 legend at upper right. The blue circle is labeled "August 1999" but it should be August 2014 as in the figure caption (line 90).

Lines 101-107 and 125-142. This is a description of the floe-identification algorithm. If it's the same algorithm as in Stern et al (2018b) then that should be explicitly stated; if not, no changes necessary.

Line 199. "The floe size may be taken to be any scalar representative of the floe size" – consider re-writing this.

Line 206. Delete "e.g." and change "black dotted and dashed lines" to "black, red, and blue dashed lines"

Line 210. "Fig. 2c through e" – should this be Fig. 2b–d?

Figure 3 caption
(i) Both "grey" (line 246) and "gray" (line 250) are used. Pick one spelling.
(ii) Line 248 refers to the "black dotted line" of 19 June 2014 in panel (a). To me it looks like a solid black line with black circular symbols. See also the legend in the lower left corner of (a).
(iii) Line 255, change sites' to site's

Line 286. "*m* appears to generally shoal with distance to the ice edge" – does this mean with INCREASING distance to the ice edge or with DECREASING distance to the ice edge?

Line 319. "conclude" should be "concluded"

Lines 347-348. This sentence repeats what was just said two sentences earlier at lines 342-343. It is redundant.

=====

[See comment above about lines 303-307]

**Notes on the size distribution of floe diameters vs. floe areas**

Let X be a random variable of floe diameters, and let *x* be a value drawn from X.
Let A be a random variable of floe areas, and let *a* be a value drawn from A.

Let $F_X(x)$ be the cumulative distribution function of X.
Let $F_A(a)$ be the cumulative distribution function of A.

Let $P\{\cdot\}$ denote the probability of the expression inside the braces.
Suppose A and X are related by $A = kX^2$ where k is a constant, and suppose $a = kx^2$. Then:

$$F_X(x) = P\{X \le x\} = P\{X \le (a/k)^{1/2}\} = P\{kX^2 \le a\} = P\{A \le a\} = F_A(a) \tag{1}$$

The probability density function (pdf) is the derivative of the cumulative distribution function.
Let $f_X(x) = dF_X/dx$ be the pdf of floe diameters.
Let $f_A(a) = dF_A/da$ be the pdf of floe areas.

Take the derivative with respect to $x$ of both sides of equation (1) and apply the chain rule of differentiation to the right-hand side:

$dF_X/dx = (dF_A/da)(da/dx)$     or:

$$f_X(x) = f_A(a)(2kx) \tag{2}$$

*Power-law pdf*

Suppose the pdf of floe area follows a power law of the form $f_A(a) = ca^m$ where $m$ is the power-law exponent and c is a normalizing constant.
Substituting this into equation (2) along with $a = kx^2$ gives:

$$f_X(x) = c(kx^2)^m(2kx) = (2ck^{m+1})x^{2m+1} \tag{3}$$

which shows that the pdf of floe diameter follows a power law with exponent $2m+1$ and normalizing constant $c' = 2ck^{m+1}$.

*Normalizing constants*

Let $a_{min}$ be the smallest floe area. Then the normalizing constant c is determined from:

$$\int_{a_{min}}^{\infty} ca^m \, da = 1 \quad \text{so that} \quad c = -(m+1)(a_{min})^{-(m+1)} \tag{4}$$

where $m+1 < 0$ in order for the integral to be finite. Let $x_{min}$ be the smallest floe diameter, and suppose $a_{min} = k(x_{min})^2$. Then the normalizing constant for the pdf of floe diameter is:

$$c' = 2ck^{m+1} = -2(m+1)(x_{min})^{-2(m+1)} \tag{5}$$

*Cumulative distribution functions*

The cumulative distribution function of floe area is:

$$F_A(a) = \int_{a_{min}}^{a} c(a')^m \, da' = \left(\frac{c}{m+1}\right)(a^{m+1} - a_{min}^{m+1}) = 1 - (a/a_{min})^{m+1} \tag{6}$$

The cumulative distribution function of floe diameter is:

$$F_X(x) = 1 - (x/x_{min})^{2(m+1)} \tag{7}$$

Neither $F_A(a)$ nor $F_X(x)$ is a power-law distribution. However, the *complementary* cumulative distributions $F'_A$ and $F'_X$ are power laws:

$$F'_A(a) \equiv 1 - F_A(a) = (a/a_{min})^{m+1} \quad \text{and} \quad F'_X(x) \equiv 1 - F_X(x) = (x/x_{min})^{2(m+1)} \tag{8}$$

The *complementary* cumulative distributions are used in the sea-ice floe-size literature. The power-law exponent of $F'_X$ is twice the power-law exponent of $F'_A$. The pdfs are now related to $F'_X$ and $F'_A$ by $f_X(x) = -dF'_X/dx$ and $f_A(a) = -dF'_A/da$ (note the minus signs).

*Finite upper limit*

Suppose the pdf of floe area is a power law, $f_A(a) \sim a^m$, but the largest floe area is $a_{max} < \infty$. Then the cumulative distributions $F_A$ and $F'_A$ are:

$$F_A(a) = \frac{1-(\frac{a}{a_{min}})^{m+1}}{1-(\frac{a_{max}}{a_{min}})^{m+1}} \quad \text{and} \quad F'_A(a) = \frac{(\frac{a}{a_{min}})^{m+1}-(\frac{a_{max}}{a_{min}})^{m+1}}{1-(\frac{a_{max}}{a_{min}})^{m+1}} \tag{9}$$

neither of which is a power law. If $a_{max} = k(x_{max})^2$ where $x_{max}$ is the largest floe diameter then the expressions for $F_X(x)$ and $F'_X(x)$ are the same as in (9) but with exponents $2(m+1)$ instead of $m+1$ and with $x_{min}$ and $x_{max}$ replacing $a_{min}$ and $a_{max}$. Again, neither $F_X(x)$ nor $F'_X(x)$ is a power law. Their derivatives (the pdfs) are power laws between the bounds $x_{min}$ and $x_{max}$ (or $a_{min}$ and $a_{max}$).

---

## Author Comment (AC1)

**Author Response to:**

Review for *The Cryosphere* of

Characterizing the Sea-Ice Floe Size Distribution in the Canada Basin from High-Resolution Optical Satellite Imagery

by Denton and Timmermans

Author Comments are in blue.

The authors analyzed 78 images from the Canada Basin of the Arctic Ocean. The images have 1-meter spatial resolution and span the period 1999-2014 during the months April to September. They identified individual ice floes in the images and analyzed the floe size distribution (FSD), where size is measured by floe area. They have four main results: (1) The FSDs follow a power-law distribution between areas $5 \times 10^1$ m$^2$ and $5 \times 10^6$ m$^2$ (50 m$^2$ to 5 km$^2$) with power-law exponents ranging from $-1.65$ to $-2.03$. (2) The FSDs are sensitive to the threshold used to separate ice from water in the images. Other studies may have erroneously found two power-law regimes by not setting the proper threshold between ice and water. (3) A linear relationship is found between power-law exponents and sea-ice concentration (SIC), with more negative exponents corresponding to lower SIC. (4) Locations that experience a seasonal cycle in SIC also have a seasonal variation in power-law exponent, but sites with high year-round SIC do not.

The analysis and conclusions of this paper are generally sound. I recommend publication after the authors consider the following comments and suggestions, which are given in page order.

Thank you for your thoughtful review with many valuable suggestions. We have taken into consideration each of your comments to improve the manuscript and our responses are inline below.

Comments and suggestions

Lines 9-11. "the structure of the FSD is found to be sensitive to a classification threshold value… and an objective approach to minimize this sensitivity is presented." I searched throughout the paper for the objective approach, but all I could find was this (on lines 163-164): "we iteratively increase the threshold above the minimum until the edges of small floes are appropriately delineated." I have no objection to this method, but I wouldn't call it an objective approach that minimizes anything. It sounds like "visual inspection" or "manual selection" to me. If that's what it is, please say so. If I'm missing the objective approach, please provide more detail.

Thank you. Our use of "objective" was referring to a choice that uses the histogram of pixel grayscale values, but we understand your point. We have altered the text on these lines to "an approach to account for this sensitivity is presented". We have also deleted the word "objective" in Sect. 2.2.

Lines 182-189. This describes the construction of the FSD by binning the data and fitting a line (in log-log space) to the binned values. That's fine, but it's a shame that the authors did not use Maximum Likelihood Estimation to find the best-fitting exponent, which does not require binning the data (and hence avoids the problem of having an adequate number of samples per bin), and which gives a more accurate estimate of the exponent than least-squares fitting of binned data. No changes necessary, I just wanted to bring up this point.

Thank you for your comment. We now also use the maximum likelihood estimator (MLE) to compute slopes; please see our response to Reviewer 2 with respect to this. We have found that the MLE returns slope values $m_{MLE}$ that differ (on average) from our least-squares fit slope values $m$ by about 3%. The mean $m_{MLE}$ over our 78 images is $-1.77 \pm 0.11$, while mean $m$ is $-1.79 \pm 0.08$ (where uncertainty bounds represent the standard deviation). We now include slopes $m_{MLE}$ in Table A1 and have added the following sentences to the manuscript in Sect. 3.1: "We find no significant difference between slopes $m$ and $m_{MLE}$ (Table A1). The mean $m_{MLE}$ over all images is $-1.77 \pm 0.11$. Considering each image, $m_{MLE}$ differs from $m$ by about 3% on average".

Lines 199-203. The authors choose to use floe area as the measure of floe size, because once the pixels of a floe have been identified, it's a simple matter to count them and multiply by the area per pixel. However, it's also a simple matter to directly calculate the mean caliper diameter (MCD) from the coordinates of the floe pixels. I'm not talking about an approximation that relates the average MCD to the average area (as in Rothrock and Thorndike, 1984) – I'm saying that the MCD can be easily calculated exactly for every individual floe, as follows.

In a pixel-based coordinate system with (0,0) at the lower left corner of the image, let $(x_k,y_k)$ be the pixel coordinates that comprise a floe, denoted by vectors x and y. Then:
for angle = 0, 179 do begin          ; Loop over caliper angles from 0° to 179° in 1° increments
   c = cos(angle * pi / 180.)
   s = sin(angle * pi / 180.)
   **r** = c\***x** + s\***y**
   diameter(angle) = max(**r**) – min(**r**) ; This is the caliper diameter at the given angle
endfor
MCD = total(diameter)/180.          ; Average over all angles of the calipers to get MCD

There's nothing wrong with using floe area as the measure of floe size, but in 18 previous studies of the FSD, 17 of them have used a diameter or dimensional length, and only one has used area. It would be easier to compare the results of this study with previous results if the authors had used MCD or another length scale. No changes necessary, I just wanted to bring up this point.

Thank you for providing the useful algorithm. We chose floe size to be represented by area as it is most directly relatable to floe models and the most computationally efficient in our set-up. Fortunately, the approximate relationship between mean caliper diameter (MCD) and floe area (Rothrock & Thorndike, 1984) works well. We have added the following phrase to a sentence in Sect 2.3: "In the present work, we use floe area because we obtain this directly in the segmentation (and it is directly relatable to floe models),..."

Lines 216-218. "the slope of the FSD is valid only for floe areas larger than 50 m$^2$ and we limit fitting in log-log plots (to estimate $m$) to floe areas between 50 m$^2$ and 5 km$^2$" How many floes fall outside these bounds, roughly? Is it 10% of all floes? 50%? Table A1 gives the "number of floes (whole) retrieved" (according to the caption on page 16). Does the number in the table include only the floes between the given bounds, or all floes? If it's all floes, then another column should be included in the table giving the number of floes with area between the bounds.

The number of floes in Table A1 includes all floes, and we now make this clear. Following your suggestion, we have added a column to Table A1 of the number of floes which are within the slope-fitting bounds (50 m$^2$ to 5 km$^2$). This value is between 10-55% of the total number of floes in the segmentation, and we have added the following sentence to Sect 2.3.1: "The number of floes which fall in this fitting range (see Table A1) is between 10 to 55 percent of the total number of whole floes segmented across the entire floe range between 5 m$^2$ and 100 km$^2$ (with the minimum floe count requirement of 2 per bin)".

Line 240 and Figure 3 caption. Some of the images are segmented (divided into ice and water) with high confidence and others with low confidence. Table A1 should indicate which images are segmented with high confidence and which with low confidence.

We now include a column in Table A1 indicating low and high confidence image segmentations.

Figure 3 panels (b) and (c) include "error bars representing one standard deviation" (lines 252 and 254). However, in some cases the number of samples is 1, so it makes no sense to calculate a standard deviation (which is 0) and call it an error bar. The number of samples for a given month and location never exceeds 7, and in most cases it's 1, 2, or 3. It's of dubious statistical rigor and utility to calculate standard deviations and call them error bars in these cases. Furthermore, the error bars make the plots more difficult to read. In my opinion, all the error bars should be removed from panels (b) and (c), and reference to them deleted from the caption.

Thank you for your comment. We have removed the monthly standard deviations as error bars and modified the caption accordingly; individual errors plotted on each slope value were too difficult to see on the plot, but these uncertainties are reported in Table A1.

Figure 3 panels (b) and (c) also include "mean monthly slopes" and "mean monthly SICs." In my comment below about Table A1, I note that not all the images provide independent estimates of sea-ice properties because some images were acquired on the same day of the same year at essentially the same location. For example, images #39, #40, and #41 are all from 20 May 2013 at the Northern Canada Basin site (see Table A1). Therefore, in calculating mean monthly slopes for May, the values from #39, #40, and #41 should be averaged first, and then their average should be averaged with the other May values from the same site (i.e., images #2, #13, #37, and #63). Perhaps the authors have actually calculated the means in this way. If not, I'm sure it wouldn't make much difference in the final results, but I just wanted to bring up the point about statistical non-independence of the images.

Thank you, this is a good point and we had not originally computed averages to account for this. We have amended our calculation of the monthly mean slopes and SICs at the three GFL sites

(Figure 3b and c), to be computed from all samples in a month after averaging any non-independent sample values acquired on the same day and site, and amended the panels to represent these newly computed means (there is negligible difference between these newly calculated means and our previous monthly means). In addition, we have added the following sentence to the end of Sect. 3.2: "Note that for each of the GFL sites, we compute mean monthly slopes after first taking the mean of any slopes from images acquired on the same day at a particular site, to account for the fact that these are not independent estimates (see Table 1). We do the same for mean monthly SIC at the GFL sites, discussed in the next section."

Lines 290-299. This paragraph is about the lack of a relationship between surface air temperature (SAT) and power-law exponent m of the FSD. The lack of a relationship is not surprising. During the Arctic summer, the SAT over ice is pegged at the melting point of ice because the surface is an ice bath (ice and water). For example, see the plots of SAT in Rigor et al. (2000). From the time of melt onset to the time of freeze-up (roughly June through August in the Beaufort Sea) there is essentially no variability in the SAT over ice, so it can't possibly explain the variability in FSD. The solar energy that goes into the ocean melts ice, potentially changing the FSD, but it does not raise the SAT over ice. In my opinion, it would have made more sense to look for a connection between m (FSD) and some other thermodynamic variable that characterizes ocean heat content or energy balance.
Rigor, I.G., R.L. Colony, and S. Martin (2000). Variations in Surface Air Temperature Observations in the Arctic, 1979–97. J. Climate, pp. 896-914, https://doi.org/10.1175/1520-0442(2000)013<0896:VISATO>2.0.CO;2

We agree that the range in SATs (here we consider the 2-m value) in summer may not be the most appropriate indicator to characterize the state of the pack because the presence of sea ice (at least when the concentration is sufficiently large) ties the 2-m SAT to a fairly small range. Because SAT is the most accessible and reliable external parameter to characterize the system, we use it here more as an indicator of when we might expect sea-ice melt, both at the atmosphere-ice and ice-ocean boundaries (i.e., when SATs fall approximately above 0°C). We agree that the relationship between FSD and surface-ocean heat content (or ocean-to-ice heat fluxes) would be more insightful, and we now refer explicitly to ocean-to-ice heat fluxes in the manuscript. Although the concurrent data in this regard are limited, a small subset of case studies where this relationship can be reliably examined would be a valuable follow-up study.

Lines 303-307. The authors state that a power-law probability density function (pdf) of the form $n(a) \sim a^m$ where $a$ is floe area is equivalent to a power-law pdf of the form $x^{2m+1}$ where $x$ is floe diameter. This is correct, but the reason given is wrong. It has nothing to do with the binning of data or the normalization of bin counts. It follows from a straightforward application of basic probability theory – see the notes at the end of this review. Lines 303-307 should be re-written to simply state that $a^m$ is equivalent to $x^{2m+1}$ as a result of basic probability theory.

Thank you for your comment and notes. We now state in these lines that: "From an application of basic probability theory, slopes reported in studies that examine the non-cumulative FSD using normalized floe number densities constructed from $x$ are equivalent to $2m + 1$ (where $m$ refers to slopes found in this study)."

Line 315. "image not included in our analysis due to partial cloud-cover" Somewhere in the paper the authors should state how many images were rejected due to partial or total cloud cover, and how those decisions were made. Was it by visual inspection?

Generally, the cloud covering (partial or full) is unambiguous in the Medea images and it is visually straightforward to decide to reject an image from analysis. We do not note how many images were rejected due to cloud cover, as there were many on the GFL which clearly indicated cloud cover and were rejected outright and not included in data download. See for example the image below taken on 22 May 2012 at the Chukchi Sea GFL site.

[Figure]

Cloud-covered Medea image acquired at the Chukchi Sea Global Fiducial Site on 22 May 2012. (Image from the United States Geological Survey Global Fiducials Library).

We have added the following sentence in Section 2.1, paragraph 1: "We note that partially or fully cloud-covered images on the GFL were generally unambiguous and rejected outright from our analysis. Cloudy pixels either fully obscure information about the ice cover below or interfere with the proper identification of floe outlines."

Lines 346-350. "seasonal variation in m is more directly related to changes in SIC" (than SAT). Yes, but as noted above, there's no reason to expect a connection between m and SAT. "Future studies are needed to investigate the relevant dynamics … and thermodynamics…" Yes, undoubtedly the same forces that drive changes in SIC also drive changes in FSD. This raises the question: why try to relate SIC and FSD in the first place? Neither one drives the other, they're both the result of underlying dynamic and thermodynamic forcing. It seems like SIC could never be more than an imperfect reflection of FSD. Imagine an image with a power-law FSD and a certain value of SIC. Now double the size of the image by adding only ocean pixels. The FSD remains exactly the same, but the SIC is cut in half. Conversely, it's easy to imagine a scenario

in which the power-law exponent of the FSD changes but the SIC does not. What is the motivation for relating FSD to SIC? Wouldn't it make more sense to look for connections between FSD and, say, wind stress?

Thank you for the comment. We were motivated to explore the FSD in context with SIC because a relationship between the two leads to follow-up scientific questions along the lines of what common dynamic/thermodynamic forcing might be responsible, and whether this relationship differs across seasons or regions. We do not see evidence for an independence like in the example case you give (SIC cut in half while FSD stays the same), although we could not rule that out as being a factor. The other case you note (varying FSD in the absence of varying SIC) would be a useful result. There are cases, for example, where the FSD slope could steepen (fewer large floes, more smaller floes) while the SIC remains the same, and this might indicate a fracturing. Conversely, a shoaling of the FSD slope associated with loss of small floes (e.g., via relatively rapid lateral melt of smaller floes compared to larger floes) may be associated with a different SIC-FSD relationship than some change to the floe field as a result of a wind event, for example. Our follow-up work is examining these effects (and as you mention, the relationship between FSD and wind stress).

We have added the following sentences to Sect. 4: "Future studies are needed to investigate the relevant dynamics (i.e., wind-forced sea-ice deformation and breakup) and thermodynamics (e.g., ocean-to-ice heat fluxes) of the sea-ice pack to explore the precise mechanisms by which the sea-ice concentration relates to the structure of the FSD, and how this relationship might differ in different settings. For example, a scenario might be envisioned where the FSD slope could steepen (e.g., as a result of fewer large floes, and more smaller floes) while the SIC remains the same, and this might indicate a fracturing. Conversely, a shoaling of the FSD slope associated with the loss of small floes (e.g., via relatively rapid lateral melt of smaller floes compared to larger floes) may be associated with a different SIC-FSD relationship."

Table A1. This is a good and useful table, but it is incomplete. As noted above, it should include the number of floes with area between 50 $m^2$ and 5 $km^2$ (i.e., the bounds used in determining the power-law exponent m) as well as the total number of floes (which is given already, I believe).

We have included a column in Table A1 which provides the number of floes with area between 50 $m^2$ and 5 $km^2$.

Also, the table should indicate which images were segmented with high confidence and which with low confidence.

We have included a column in Table A1 which indicates high and low confidence segmentations.

Also, there should be a column to indicate whether an image is from the Beaufort fiducial site, the Chukchi fiducial site, the Northern Canada Basin fiducial site, or another site. I realize that this can be inferred from the latitude and longitude, but it is extremely tedious to go through the table and extract that information.

Thank you for pointing this out. We have included this information in Table A1.

Another point that should be noted somewhere in the paper is that some of the images were acquired on the same day of the same year at essentially the same location, and therefore do not provide independent estimates of sea-ice properties (FSD and SIC). In particular:
**3 and #4 are both 27 July 2000 at the N. Basin site**
**5 and #6 are both 15 Aug 2000 at the N. Basin site**
**14 and #15 are both 23 May 2002 at the Beaufort site**
**39 and #40 and #41 are all 20 May 2013 at the N. Basin site**
**43 and #44 are both 10 June 2013 at the Chukchi site**
**45 and #46 are both 12 June 2013 at the Beaufort site**
**57 and #58 are both 28 April 2014 at the Beaufort site**

We have recomputed the means and noted this in the text (see our earlier response).

I found it useful to summarize the number of images by location and month in a table. Consider including something like this in the paper:

|  | Beaufort | Chukchi | N. Basin | Other | Total |
|---|---|---|---|---|---|
| **April** | 5 (*) | 2 | 1 | 17 | 25 |
| **May** | 7 (*) | 2 | 7 (*) | 4 | 20 |
| **June** | 2 (*) | 5 (*) | 1 | 7 | 15 |
| **July** | 1 | 1 | 3 (*) | 3 | 8 |
| **Aug.** | 1 | 0 | 3 (*) | 2 | 6 |
| **Sept.** | 1 | 0 | 1 | 2 | 4 |
| **Total** | 17 | 10 | 16 | 35 | 78 |

The (*) symbol indicates that some of the images are from the same day of the same year at essentially the same location, i.e., not independent samples.
**Beaufort** site is shown in Figure 1(a) at the letter e.
**Chukchi** site is shown in Figure 1(a) at the letter c.
**N. Basin** site is shown in Figure 1(a) at the letter d.
**Other** sites are shown in Figure 1(a) at white dots.

Thank you for your suggestion. We have included a Table 1 in Sect 2.1 with the number of images (per month and total) at each GFL site.

Minor typographical notes

Line 6. Change "atmosphere and ocean" to "atmospheric and oceanic" as on line 22.

We have amended the text accordingly.

Line 64. "Aeronautical" should be "Aeronautics"

Thank you. We have altered the text accordingly.

Line 66. "The images AT FIDUCIAL SITES are panchromatic…"

All images used in our study are panchromatic, not just images at fiducial sites.

Last sentence on page 3. Please say that the surface air temperature (SAT) is at the 2-meter level.

We have amended the text accordingly.

Figure 1 legend at upper right. The blue circle is labeled "August 1999" but it should be August 2014 as in the figure caption (line 90).

Thank you for catching this typo; we have corrected it. We have also corrected the location of e on the map and added asterisks to more clearly delineate GFL fixed sites. We have altered a sentence in the figure caption: "USGS fiducial sites (black asterisks), for which there are images from multiple years, are noted to the southeast of e (Beaufort Sea), at c (Chukchi Sea), and at d (northern Canada Basin)".

Lines 101-107 and 125-142. This is a description of the floe-identification algorithm. If it's the same algorithm as in Stern et al (2018b) then that should be explicitly stated; if not, no changes necessary.

Thank you for your comment. The algorithm is influenced by multiple studies' floe segmentation approaches which are referenced in the text. While inspired by and similar to Stern et al. (2018b), and specifically including Stern et al. (2018b)'s hierarchical segmentation scheme into our segmentation algorithm (as is referenced), it is not clear based on the description by Stern et al. (2018b) whether the algorithm is exactly the same. For instance, we do not use the morphological dilation operation from Serra (1982) to expand floes to their original size, as referenced by Stern et al. (2018b), but rather a rule-based expansion using the numerical mode of the floe label neighborhood as presented by Paget et al. (2001). Stern et al. (2018b) indicate that their algorithm is "similar to that of Paget et al. (2001)", but not that it is identical. We therefore have not made any changes to our description except for the addition of a reference to Stern in Sect. 2.2 under "Erosion and Expansion".

Line 199. "The floe size may be taken to be any scalar representative of the floe size" – consider re-writing this.

We have amended the text to read "The floe size may be taken to be floe area $a$, perimeter, or a diameter proxy such as the mean caliper diameter (MCD), …".

Line 206. Delete "e.g." and change "black dotted and dashed lines" to "black, red, and blue dashed lines"

We have amended the text accordingly.

Line 210. "Fig. 2c through e" – should this be Fig. 2b–d?

Thank you. We have amended the text accordingly.

Figure 3 caption
(i) Both "grey" (line 246) and "gray" (line 250) are used. Pick one spelling.

We have altered the text to use the American English spelling of the color, "gray".

(ii) Line 248 refers to the "black dotted line" of 19 June 2014 in panel (a). To me it looks like a solid black line with black circular symbols. See also the legend in the lower left corner of (a).

We have amended the caption text accordingly.

(iii) Line 255, change sites' to site's

Thank you for noting this; we have amended the text accordingly.

Line 286. "$m$ appears to generally shoal with distance to the ice edge" – does this mean with INCREASING distance to the ice edge or with DECREASING distance to the ice edge?

We have altered the text to read "$m$ appears to generally shoal with increasing distance from the ice edge".

Line 319. "conclude" should be "concluded"

We have altered the text accordingly.

Lines 347-348. This sentence repeats what was just said two sentences earlier at lines 342-343. It is redundant.

We have deleted the corresponding sentence on lines 347-348.

=====

---

## Author Comment (AC2)

**Author Response to:**

Christopher Horvat

This is my review of Denton and Timmermans (2021).

Author Comments are in blue.

Here the authors use image processing to obtain floe size distribution (FSD) measurements spanning a scale range from 50 m^2 -5 km^2, and over a wide time period in the Canada Basin. They find both that FSDs obey a power law relationship and that there is a connection between sea ice concentration and power law coefficient. Generally I find this paper to be an interesting contribution to the literature. It is well written and has a comprehensive review of existing work and does a good job of contextualizing the work here. I am also particularly excited that someone has characterized the GFL dataset! I have a few minor comments, and one important one about statistics that needs to be taken into account. The latter is a straightforward application of a method to data already presented in the MS, so it really is likely to be a minor revision, though it may have an impact on the results. So for the authors I'm recommending "major" revisions only to give more time to include that bit of analysis and discussion. Up to these changes, I'm happy to recommend this paper for publication.

Best,

Chris Horvat

\_\_\_\_

Thank you, we appreciate your helpful review and overall suggestions related to power-law fitting statistics. We have taken into account each of your comments (inline below) to improve the manuscript.

Main comment.

L30, and generally - It is worth describing the power-law behavior of the FSD as a "hypothesis". As was pointed out by Herman (2010), there actually isn't strong evidence of a power law in most data, but the use of this to describe the distribution is because it is handy for multi-scale distributions.

Thank you for this comment. We now reference this paper and add the following clause to the sentence on lines 35-38 : "...although alternate distributions have been explored (see e.g., Herman, 2010, and the discussion by Stern et al., 2018a)."

As was pointed out in Stern's two 2018 papers, the appearance of power-law behavior can appear spuriously as a result of choosing a particular way of plotting the distribution. This hypothesis can be formally tested, and the mechanism for doing so is described in papers by Clauset (2007,2009,2014), who maintains code to perform fitting, goodness-of-fit tests, and p-value computations at https://aaronclauset.github.io/powerlaws/. When examining multi-scale distributions, one should report the p-value for fits, and the start of a power-law tail. There are many reasons for this!

Thank you, we now report $p$-values and the strict statistical lower-bound on power-law behavior for the power-law fits, $a_{min}$, following the methods of Clauset et al. (2009) and employing Aaron Clauset's power-law MATLAB toolbox. Please see our later responses.

First, often the intuitive "eye test" for power-law behavior is flawed, and alternative distributions fit better (as examined by Herman (2010)). In studies where this was adamantly required by certain unnamed co-authors, (e.g. Hwang, 2017), few of the obtained distributions actually had power law tails, or only could not reject a power law hypothesis over a small range of size scales. It is worth knowing this!

Thank you for your comment, which we address below. We note that Hwang et al. (2017) report that 77% of their data passed the goodness-of-fit test ($p$-value $\geq 0.1$). Similarly, Stern et al. (2018b) find that "...only 9 out of 116 floe size data sets are rejected as not being power-law distributed in the 2013 MODIS imagery, and only 3 out of 140 floe size data sets are rejected in the 2014 MODIS imagery." They conclude that "in general the FSD in the Beaufort and Chukchi seas follows a power-law distribution for floes from 2 to 30 km in size." Neither of these groups elaborate on the statistically tested range of size scales over which a power law holds.

Second, power-law behavior is fundamentally about the tail of a distribution. Thus issues at the "artifact scale" are not important when performing such tests. This allows you to expand the range of floes into the less certain smaller sizes without actually losing information - if the fit doesn't extend there, it won't be included.

Thank you for the comment. In our case, it is clear a-priori that the floes at or below the "artifact scale" should not be considered in the least-squares fit. We discuss our computation of FSD slopes using MLE in the next response. For those fits, we do not exclude the artifact scale, but we do limit the data to floe sizes larger than the image resolution-limit scale (5 m$^2$).

Third, there is a mechanism for looking at binned (Virkar and Clauset, 2014) and un-binned data (Clauset et al, 2007,2009). The binning employed here seems to be done to facilitate fitting, but this is not necessary when using the Clauset-style tests. You have the full information on floe areas, so you can directly use that to obtain fits, goodness of fits, and p-values.

We now determine the MLE power-law fit, $m_{MLE}$, the strict statistical lower-bound on power-law behavior for the MLE fits, $a_{min}$, and goodness-of-fit for these MLE power-law models indicated

by corresponding *p*-values, from our unbinned floe areas (for all images) following Clauset et al. (2009) and using Clauset's power-law MATLAB toolbox (https://aaronclauset.github.io/powerlaws/). We have added the following section 2.3.2 to the manuscript:

"The maximum likelihood estimator (MLE, see Clauset et al., 2009) can be preferable for the determination of FSD slopes as it does not rely on specifying bins or fitting ranges (Hwang et al., 2017; Stern et al., 2018a; Stern et al., 2018b). In addition to least-squares fitted slopes *m* and following Clauset et al. (2009), we compute FSD MLE slopes $m_{MLE}$, and conduct goodness-of-fit tests on these power-law fits, reporting corresponding *p*-values (where the *p*-value is the probability that the difference between the model fit and the observed FSD could be due to statistical fluctuations; see Clauset et al., 2009 for a detailed discussion), Table A1. The power-law fit is a plausible model for the FSD if the computed *p*-value is sufficiently large ($p \geq 0.1$); otherwise, the power-law model must be rejected.

"Clauset et al. (2009) argue that a strict statistical lower bound on power-law behavior must be computed for the observed distribution; we compute these values $a_{min}$, following their methodology (Table A1). Because $m_{MLE}$ and $a_{min}$ are determined directly from the unbinned floe areas for each image, we compute both over all floe areas ($\geq 5$ m$^2$), and do not exclude floes at or below the artifact scale."

We find that:

1. The resulting non-cumulative FSD slopes using Clauset's unbinned MLE method, $m_{MLE}$, are not significantly different than our least-squares fitted slopes, *m*, determined over our fitting range (50 m$^2$ to 5 km$^2$). For reference, the mean $m_{MLE}$ over our 78 images is -1.77 ± 0.11, while mean *m* is -1.79 ± 0.08 (where uncertainty bounds represent the standard deviation); considering each image, $m_{MLE}$ differs from *m* by about 3% on average. We now include slopes $m_{MLE}$ in Table A1 and have added the following sentences to the manuscript in Sect. 3.1: "We find no significant difference between slopes *m* and $m_{MLE}$ (Table A1). The mean $m_{MLE}$ over all images is -1.77 ± 0.11. Considering each image, $m_{MLE}$ differs from *m* by about 3% on average".

2. In addition, following Clauset et al. (2009), we find that for all 78 images, $a_{min}$ spans 11 m$^2$ to 9,808 m$^2$. We report these in Table A1. We note that for our dataset, the median value of $a_{min}$ is 361 m$^2$ while the largest floe areas are around 10 to 100 km$^2$. Even considering the maximum value of $a_{min}$, this is a large range of floe sizes over which the power-law models are plausible. We add the following sentences to Sect 3.1, "Finally, we find that the strict lower-bound to power-law behavior $a_{min}$ varies considerably over the images (Table A1), spanning around 10 to 10,000 m$^2$ with a median value of $a_{min}$ of 361 m$^2$. Considering that the largest floe areas in the images are around 10 to 100 km$^2$, the range of floe sizes over which the power-law fits apply is large. Values of $a_{min}$ can vary significantly even across images acquired on the same day at the same location (see e.g. Table A1, images 14–15)."

3.  We conduct goodness-of-fit tests using the K-S statistic (following Clauset et al., 2009) on the MLE-fitted power-law models (power-law fits with slopes $m_{MLE}$ and lower-bound to the fits $a_{min}$) to our 78 FSDs, finding that 76% of them pass the test ($p$-value $\geq 0.1$), indicating that the power-law fit is a plausible model to our floe size data. We have added the corresponding column of $p$-values for each image segmentation to Table A1 and the following sentence to Sect. 3.1: "We find that 76% of the fits pass the goodness-of-fit test with $p \geq 0.1$ (Table A1) meaning that the FSDs can plausibly be power-law distributed."

I would suggest that these tools be applied to the distributions obtained here, and p-values and tail beginnings (x_{min} in the power law toolbox) reported for power-law fits before making statements about the distributional fitting. Without them, it is hard to be convinced rigorously that this hypothesis is not "not true". Including these makes the reporting of PL behavior robust.

Please see our above response to your comments regarding the power-law fits, tail beginnings, and $p$-values.

Finally, it would be good for the community if your segmentation algorithm was posted publicly and DOId, through e.g. Github/Zenodo. Almost all existing algorithms used in FSD studies are similar in structure, but not able to be used by others. Thus each time a new group wants to compute FSDs they have to reinvent the wheel. Having an available FSD code would benefit many, and ensure your work was properly credited.

We appreciate your suggestion and will share the segmentation algorithm on such a repository. We will note in the final comment period where we have deposited the algorithm.

Specific Comments:

L26 - it is worth looking at the papers of Roach (2018,2019), Zhang et al (2017) who also discussed and theorized the impact of the FSD on both local and model-scale melt partitioning.

Thank you. We now cite these papers in Sect 1.1.

L95 - can you clarify how your method differs from the methodology used by other authors? Is it a similar concept?

Please see our response to Reviewer 1 in this regard.

L198 - I appreciate that you are using floe area to define the FSD - not sure why more authors don't do this. It makes for more accurate comparison to models (which use a fixed radius-area relationship) and is much easier to understand than the MCD. I would even add a statement here or in the discussion to make this point as it is quite helpful from the modeling side.

Thank you for your comment. See our response to Reviewer 1. We have added the following phrase to a sentence in Sect. 2.3: "In the present work, we use floe area because we obtain this directly in the segmentation (and it is directly relatable to floe models)..."

L180 - Have a look at the main comment here. I do not think it is a good idea to compute power law slopes from binned data, as you are sensitive to your bin choices. Still, there is a methodology for computing such slopes (Virkar and Clauset, 2014), which I would argue to employ here if you want to compute them using the binned data. However you have the raw area data, and so don't need to resort to binning. In that case you can use the methodology of Clauset et al (2009) to obtain slopes and p-values for fitting. You will find that the binned PL slopes and the unbanned PL slopes *will* differ.

Thank you. We have done this now. Please see our comments above.

L223 - Again, I would strongly caution to use the Clauset method to obtain slopes. This is discussed in the series of Clauset papers, and later by Stern et al (2018), that fitting straight lines to log-log plots can often fail you in unexpected ways.

Please see our comments above.

L279 - please clarify how you are testing for significance. You are highly sampled in the high SIC range, and weakly so in the low SIC range. What went into the choice of such a significance test, and why? For example, you bin floe areas and then fit them - what guides that choice, but not a similar choice for SIC? Hopefully that points a bit towards why it might be preferable to avoid binning.

Thank you for your comment. We note that the $p$-value on the linear fit between SIC and $m$ is $3.18 \times 10^{-8}$ (with the null hypothesis being that there is no relationship between the two), below the significance level 0.01, suggesting that the null hypothesis may be rejected, and the two variables share a statistically significant relationship. The r-squared value of our fit is 0.33, indicating that there is a relatively large degree of variability in $m$ with SIC. However, the statistically significant relationship between the two remains. With regard to nonuniform sampling, we note that a residual plot of the linear fit between SIC and $m$ indicates no clear bias or pattern to the residuals, which indicates that a linear model can be appropriate for the data here.

We have added the following sentence to Sect. 3.3: "Note that there are more sample points in the high SIC range than in the low range, and the linear fit can only explain 33% of the variation (r-squared is 0.33) in $m$ with SIC. However, the linear relationship is statistically significant with a $p$-value of $O(10^{-8})$ (i.e., $< 0.01$)."

With regard to avoiding binning for our floe areas, please see our earlier response.

L360 - An extensive discussion of how multiple FSD size regimes can emerge was performed in Horvat and Tziperman (2017) which I will shamelessly plug. It is worth noting the difference in locations may impact the processes that give rise to PL behavior (i.e. the impact of waves leads to more small floes, etc).

Thank you for the useful reference. We have added the following sentence to Sect. 4: "For example, Horvat and Tziperman (2017) use a coupled ice-ocean model to show that increased lateral melt on specific floe sizes and transient oceanic forcing on the ice pack can perturb the FSD behavior from a single power-law at the relevant scale."

References.

Clauset, Young, and Gleditch. On the Frequency of Severe Terrorist Events. 2007

Clauset et al. Power-Law Distributions in Empirical Data. 2009.

Virkar and Clauset. Power-law distributions in binned empirical data. 2014.

Herman. Sea-ice floe-size distribution in the context of spontaneous scaling emergence in stochastic systems. 2010.

Horvat and Tziperman. The evolution of scaling laws in the sea ice floe size distribution. 2017.

Roach et al. An emergent sea ice floe size distribution in a global coupled ocean--sea ice model. 2018.

Roach et al. Advances in Modeling Interactions Between Sea Ice and Ocean Surface Waves. 2019.

Zhang et al. Sea ice floe size distribution in the marginal ice zone: Theory and numerical experiments. 2016.